# Scaling the tail beat frequency and swimming speed in underwater undulatory swimming

Jesús Sánchez-Rodríguez ●[1,2,3], Christophe Raufaste ●[1,4] & Médéric Argentina ●[1] ✉

Undulatory swimming is the predominant form of locomotion in aquatic vertebrates. A myriad of animals of different species and sizes oscillate their bodies to propel themselves in aquatic environments with swimming speed scaling as the product of the animal length by the oscillation frequency. Although frequency tuning is the primary means by which a swimmer selects its speed, there is no consensus on the mechanisms involved. In this article, we propose scaling laws for undulatory swimmers that relate oscillation frequency to length by taking into account both the biological characteristics of the muscles and the interaction of the moving swimmer with its environment. Results are supported by an extensive literature review including approximately 1200 individuals of different species, sizes and swimming environments. We highlight a crossover in size around 0.5–1 m. Below this value, the frequency can be tuned between 2–20 Hz due to biological constraints and the interplay between slow and fast muscles. Above this value, the fluid-swimmer interaction must be taken into account and the frequency is inversely proportional to the length of the animal. This approach predicts a maximum swimming speed around 5–10 m.s$^{-1}$ for large swimmers, consistent with the threshold to prevent bubble cavitation.

Beyond a few centimeters in length, most aquatic vertebrates propel themselves through the water by deforming their spines and propagating deformation waves through the body[1]. Fish, cetaceans, reptiles, amphibians, and birds oscillate the head, body, tail, forelimbs and/or fins, as appropriate, with a variety of gaits described by a specific classification[2]. Despite the complexity of treating each case separately, the kinematics of underwater undulatory swimmers can be captured to first order with a few parameters such as the wavelength of the deformation $\lambda$, the tail beat amplitude $A$ and the tail beat frequency $f$. There is now considerable evidence that $\lambda$ and $A$ are strongly related to animal length $L$, regardless of the size and shape of the animal, or the swimming conditions. In the example of fish, the wavelength

scales as the animal length, with a factor of the order of unity[3,4]. The same applies to the tail beat amplitude that follows $A \simeq 0.2L$[5–8] from tadpoles of a few centimeters to whales of 20 meters in length (see Methods). These simple allometric scaling relations reveal general physical laws that are valid over several orders of magnitude in size. Momentum balance and minimal energy expenditure associated with the hydrodynamic interaction between the moving body and the surrounding water appear to drive the selection of the amplitude, as well as the determination of the swimming speed, $U \simeq 3.3Af$[8–10]. Therefore, the swimming speed is proportional to the oscillation frequency for a given swimmer and scales as $Lf$ (see Methods), with a proportionality factor between 0.4 and 1 for fish and cetaceans[3,11–14].

[1]Université Côte d'Azur, CNRS, INPHYNI, 17 Rue Julien Lauprêtre, Nice 06200, France. [2]Departamento de Física Fundamental, Universidad Nacional de Educación a Distancia, Madrid 28040, Spain. [3]Laboratory of Fluid Mechanics and Instabilities, École Polytechnique Fédérale de Lausanne, Lausanne CH-1015, Switzerland. [4]Institut Universitaire de France (IUF), 1 Rue Descartes, Paris 75005, France. ✉e-mail: mederic.argentina@univ-cotedazur.fr

Taking the average factor, the relationship

$$U \simeq 0.7 L f \qquad (1)$$

is therefore a very good approximation of the swimming speed to within a factor of 2 at most[3,5].

The swimming speed is thus intrinsically linked to the oscillation frequency, but unlike the aforementioned scaling laws that follow clear and widely documented trends over several orders of magnitude in length, no consensus has been reached on the law that sets the tail beat frequency. Most studies agree that the frequency decreases with the length[5] and scaling laws $f \sim L^{-n}$ with an exponent $n$ ranging between 0.5 and 1 are often reported together with models referring to biological constraints, to the hydrodynamic interactions of the swimmer with its environment or even to the effect of gravity[15–19]. The difficulty of establishing a clear law and identifying the mechanisms at play is related to the fact there is only a factor of 100 between the highest frequency recorded in the smallest fish and the lowest frequency recorded in the largest cetaceans, typically 20 and 0.2 Hz respectively (Fig. 1), while measurements show a large dispersion for a given length. This dispersion has two different origins. First, by following Eq. (1), each swimmer can vary their frequency by a factor of about 10 to adjust their swimming speed[5]. Second, many factors might influence the exact frequency range such as the previous training or the condition of the animal, its age or sex, and the water or body temperature[20–22].

Obtaining experimental results that maintain homogeneity in all these features for a wide range of aquatic animal sizes is simply impossible and it is thus not surprising to find diverse experimental laws in the literature. In particular, comparing animals with the same level of activity, such as sustained, prolonged or burst, would be necessary[23] but difficult to implement in experiments[24,25]. As a consequence, instead of focusing on a specific activity gait, we propose to gather all data available in the literature, regardless of the level of activity or any other specific characteristic, to build a database of more than one thousand entries. This approach provides a complete picture of the dependency of the frequency with the length and captures both the main trend and the dispersion associated with a given length. We thus propose a frequency selection mechanism that, on the one hand, balances the swimmer's muscle force and the reactive forces generated by the fluid when the animal is in motion and, on the other hand, considers the type of muscles, slow and fast. This works uncovers allometric relations for the swimming frequency and speed that take the form of scaling laws in the limits of very small or very large swimmers.

## Results

### Tail beat frequency measurements

We collected about 1200 data points from references listed in Methods, with no discrimination on the basis of activity level or any of the other parameters mentioned above, to avoid any possible bias in the length-frequency relationship. In Fig. 1, it appears that the tail beat frequency is correlated to the length, with all measurements located within a band in the $L - f$ plane. For most of the lengths, the upper bound is well identified as the burst activity level[26–28] and the frequency varies approximately by a factor of 10 for a given $L$. For the longest animals, typically cetaceans with length $L > 5$ m, the magnitude of the band decreases. As we will discuss later, this decrease is most likely associated with a lack of burst frequency measurements: unlike the smaller animals, these animals were only observed in their natural environment and were not forced to swim at peak activity. If we define the upper and lower bounds of the band as fast and slow, we observe that both follow the same behavior: the frequency is constant and maximum at small lengths, typically $[f]_{fast} \simeq 20$ Hz and $[f]_{slow} \simeq 2$ Hz respectively, before decreasing at larger $L$. The change in tendency occurs around $L = 0.5 - 1$ m. Since the span in $f$ observed for a given $L$ is of the same order of magnitude as the span in $f$ of a given specimen to adjust its speed, we conclude that the frequency intervals are primarily associated with variations in the level of activity.

In the upcoming section, we delve into the interpretation of the dataset, taking into consideration the force-velocity relationship in muscle, the type of muscle fiber, and the mechanical interaction between the swimmer and its environment. Our approach considers the distinction between slow and fast muscle fibers to predict the slow and fast bounds of the band in the $L - f$ plane. By considering the limitations of muscle, including maximal force and frequency, for a given bound or activity level, we elucidate the transition observed around $L = 0.5 - 1$ m. These typical values mark the crossover from a regime where frequency remains relatively constant for small animals to one where frequency decreases with length for larger animals.

### Model and scaling laws

Locomotion occurs as a result of undulating movements produced by the contraction of blocks of muscle segments[29]. While the muscles contract on one side of the swimmer, those on the opposite side relax, alternately flexing the entire body from side to side. Vertebrate muscles share a large number of structural and functional features[3] that can be reasonably well described by a limited number of parameters. This is the case for the relationship between the force in the muscle, $F$, and the muscle contraction velocity, $v$, as described by Hill's muscle model[30]:

$$F = F_0 \frac{1 - \frac{v}{v_0}}{1 + \kappa \frac{v}{v_0}}, \qquad (2)$$

where $F_0$ is the maximum isometric force generated in the muscle, obtained as $v$ tends to zero, and $v_0$ is the maximum contraction velocity over which no force can be produced. In its dimensionless form, the force $F/F_0$ is a decreasing and convex

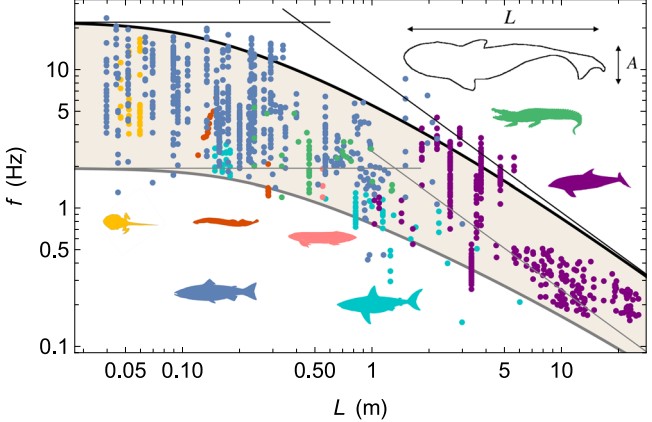

**Fig. 1 | Model predictions and observations of frequency-length.** Tail beat frequency $f$ as a function of length $L$ for agnathans (red), cartilaginous fishes (cyan), ray-finned fishes (blue), lobe-finned fishes (pink), amphibians (yellow), reptiles (green), and mammals (purple). Thick black and grey lines represent the burst and sustained activity levels, respectively, fitted with the model. Thin lines are the scaling laws in the limits of very small and very large swimmers. Values of the parameters for the fast bound are $[L_c]_{fast} = (0.4 \pm 0.2)$ m, $[f_0]_{fast} = (22 \pm 3)$ Hz, $[\kappa]_{fast} = (4 \pm 4)$, while for the slow bound we find $[L_c]_{slow} = (1.3 \pm 0.6)$ m, $[f_0]_{slow} = (1.9 \pm 0.2)$ Hz, $[\kappa]_{slow} = (14 \pm 13)$. The alabaster area represents the frequency band used by the swimmers.

function of the velocity $v/v_0$, whose degree of curvature is quantified by the parameter $\kappa$ (see Methods). Since the muscle fibers work in parallel, the maximum force scales as the cross area of the muscle: $F_0 \sim L^2\sigma_0$ where $\sigma_0$ is the maximum isometric force per unit cross-sectional area[31]. The contraction velocity $v$ drives the tail velocity and we can expect the scaling $v \sim Af$. Since the amplitude does not correlate significantly with quantities such as the swimming speed and can be considered constant for a given swimmer within the leading-order approximation[4], we can reasonably assume that $v/v_0 = f/f_0$ in what follows, with $f_0$ the maximum tail beat frequency expected from the physiological limit of the muscle. $f_0$ can be inferred by measuring the twitch contraction time $\tau$ of the muscle[32], that is, the period of a single contraction and relaxation cycle produced by an action potential within the muscle fibers, in order to deduce the maximum frequency in the form $f_0 = \frac{1}{2\tau}$ since one period consists of two antagonistic contractions.

Muscle fibers can be roughly characterized as fast or slow, in part because the latter type has a much lower level of ATP activity and a smaller contraction speed but increased activity of oxidative enzymes. Therefore, slow fibers are intended to produce forces over a prolonged period of activity[33], while fast fibers use anaerobic chemical reactions and are adapted to rapid movements, while producing higher forces. As a result, fast and slow muscles are primarily solicited in burst and sustained activity levels, respectively[34]. In what follows, we hypothesize that the presence of the lower frequency boundary of the $L-f$ graph reflects the use of slow fibers, while to reach high frequencies, fast fibers are exploited.

The swimmer's movements set the surrounding fluid in motion to propel it along. Given that the only force exerted by the animals on the fluid is provided by the muscles, there must exist a balance between the forces produced by the muscles and the reaction force of the fluid. Above a few centimeters in length, aquatic organisms have a mode of locomotion based on inertia[10]: while the body is oscillating, boluses of water of mass proportional to $\rho L^3$ are set in motion with an acceleration proportional to $Af^2$ normal to the tail, resulting in a lateral force that scales as $\rho L^3 Af^2$, where $\rho$ is the density of water. As discussed in the introduction and in the Methods section, $A$ is proportional to $L$ and thus the lateral force that pushes the fluid scales as:

$$F_{\text{lateral}} \sim \rho L^4 f^2. \tag{3}$$

By balancing the latter with the force exerted by the muscle (Eq. (2)), we obtain:

$$\left(\frac{L}{L_c}\right)^2 = \left(\frac{f_0}{f}\right)^2 \frac{1 - \frac{f}{f_0}}{1 + \kappa\frac{f}{f_0}}, \tag{4}$$

where we have introduced the length $L_c$ that marks the crossover between two regimes with different scaling laws:

$$L_c = \sqrt{\frac{\sigma_0}{\rho}}\frac{1}{f_0}, \tag{5}$$

$$f \simeq f_0, \quad \text{if} \quad L \ll L_c, \tag{6}$$

$$f \simeq f_0\frac{L_c}{L} = \frac{c}{L}, \quad \text{with} \quad c = \sqrt{\frac{\sigma_0}{\rho}}, \quad \text{if} \quad L \gg L_c. \tag{7}$$

Eq. (4) predicts nicely the two bounds of the frequency band. In Fig. 1, we have drawn the best fits (see Methods) with the set of parameters $[L_c]_{\text{fast}} = 0.4 \pm 0.2$ m, $[f_0]_{\text{fast}} = 22 \pm 3$ Hz and $[\kappa]_{\text{fast}} = 4 \pm 4$ for the fast

bound and $[L_c]_{\text{slow}} = 1.3 \pm 0.6$ m, $[f_0]_{\text{slow}} = 1.9 \pm 0.2$ Hz and $[\kappa]_{\text{slow}} = 14 \pm 13$ for the slow bound. Unlike maximum frequency $f_0$ and crossover length $L_c$, $\kappa$ has large standard deviations from the fitted values; in fact $\kappa$ weakly shapes the fit since it only plays a role in the transition between the two limit regimes. These fits are also predictive: on the example of humans and underwater undulatory swimming, also called dolphin kick, the fit of the fast bound predicts a maximum kicking frequency around 3-4 Hz for a swimmer of about 2 meters, in good agreement with data recorded in elite swimmers[35].

In addition, this model reconciles two previously proposed scaling laws and validates each in its own length range.

For $L \ll L_c$, the frequency is fixed by a biological constraint, $f_0$, in the spirit of the approach initiated by Wardle[32] who proposed that the maximum tail beat frequency corresponds to the maximum frequency expected from the muscles.

For $L \gg L_c$, the frequency decreases with length as a consequence of the interplay between another biological constraint, here the muscle stress $\sigma_0$, and the interaction of the swimmer with its environment[16,18,31]. Since the wavelength of the body deformation is of the order of the animal length in undulatory swimming[3,4], the model predicts that the frequency is given by $f = cL^{-1}$ in this limit, where the parameter $c = \sqrt{\sigma_0/\rho}$ represents the body wave speed. We estimate $[c]_{\text{slow}} \simeq 2$ m.s$^{-1}$ and $[c]_{\text{fast}} \simeq 9$ m.s$^{-1}$ for the slow and fast bounds, respectively. This scaling is obtained by approximating the swimmer by an elastic beam of length $L$, radius $R$ and Young Modulus $E$. In such a case, the natural frequency of bending waves scales is $f_e \sim \frac{R}{L^2}\sqrt{\frac{E}{\rho}}$[36]. By assuming geometrical similarity, $R \sim L$, and body elasticity compatible with stresses generated by muscle fibers $E \sim \sigma_0$, the natural bending frequency scales as the tail beat frequency $f$ of large animals, defined in Eq. (7). This suggests that large animals undulate their bodies to resonate with their natural bending modes, as evoked by various studies[37–39], even though, this hypothesis has not been validated for biological swimmers.

Finally, we now understand why there are so many different exponents in the literature resulting from attempts to describe the frequency-length relationship as a scaling law (e.g., in refs. 15–19). Special attention must be paid to the analysis of a range of lengths, because 1) the scaling laws are only valid in either of the two limiting regimes and 2) the data must be measured at the same activity level for the comparison to be meaningful.

## Considering the type of muscle fiber and comparison with biological data

In our approach, we assume that Eq. (4) is valid for a given activity level. The parameters found from the fits of the fast and slow bounds (Fig. 1) should therefore be related to the biological properties associated with fast and slow muscles respectively. Wardle et al. suggests inferring the maximum frequency $f_0$ from measurements of the twitch contraction time[32], resulting in values between 5 and 25 Hz for 4 cm to 2.3 m fish[11]. These measurements are in agreement with the value $[f_0]_{\text{fast}} = 22$ Hz that fits the fast bound. Given that the muscle stress $\sigma_0 \simeq 200$ kN.m$^{-2}$ for fast muscles is rather constant among species[40], the estimate $L_c \sim 0.7$ m is in excellent agreement with the fit $[L_c]_{\text{fast}} = 0.4$ m. For slow muscles, the maximum frequency is smaller, and we employ the same approach to study the slow bound, although measurements are rarer in this case. In the example of 10 cm salmon and 1 m sharks, $f_0$ ranges from 0.5–2 Hz, again in good agreement with the $[f_0]_{\text{slow}} = 1.9$ Hz obtained from the fit of the slow bound. Measurements of $\sigma_0$ for slow muscles are found between 20 and 80 kN.m$^{-2}$[35,40–42]. If we take 50 kN.m$^{-2}$ as the typical value, we find $L_c \sim 3.5$ m, whose order of magnitude is coherent with the value $[L_c]_{\text{slow}} = 1.3$ m found from the fit. Finally, the values of $\kappa$ adjusted for the fast and slow bounds, $[\kappa]_{\text{fast}} = 4$ and $[\kappa]_{\text{slow}} = 14$, are in the

range of values recorded in vertebrates, typically between 2.5 and 10[43,44].

This framework with six parameters, three for each bound, appears coherent because the filament arrangement in striated muscles is very similar along all vertebrates[45].

In addition to the activity level, body temperature may also play a role in influencing tail beat frequency. This is particularly evident in processes that rely on activation mechanisms[46]. For example, the twitch contraction time and subsequently, the frequency $f_0$, are affected by temperature. The impact can be significant, with a fivefold difference in $f_0$ observed between temperatures of 2 °C and 30 °C[32,47]. On the other hand, changes in body temperature seem to have only a marginal effect on $\sigma_0$[48]. In our study, we focus on $f_0$ as the primary parameter for modeling small animals with lengths much smaller than the characteristic length ($L \ll L_c$). These small animals are ectotherms (see the Methods section), and their body temperature can be associated with the ambient water temperature. In the $L < L_c$ region of the $L - f$ graph, there appears to be no correlation between tail beat frequency and water temperature. This lack of correlation is likely due to the relatively small temperature differences among individuals, typically around 15–20 °C (see the Methods section). In the case of large animals whith $L \gg L_c$, $\sigma_0$ becomes the key parameter, and the influence of body temperature on tail beat frequency is expected to be limited. It is only at intermediate lengths, approximately 3 meters, that we observe a significant difference in body temperature, such as between mammals at 37 °C and ectothermic *Greenland sharks* in Arctic waters at around 0 °C. This extreme difference in temperature likely explains the surprisingly low tail beat frequency of *Greenland sharks* (approximately $L \simeq 3$ m and $f \simeq 0.15$ Hz in Fig. 1), which is 2-3 times lower than the fit of the slow bound[17]. Apart from this particular case, assuming that the six parameters of the model remain constant across the entire length range is a reasonable first-order approximation.

### Scaling the swimming speed

In Fig. 2, we have plotted the speed data reported for natural swimmers (same references as for frequency data, see Methods). Similar to the frequency measurements, the speed measurements also exhibit a characteristic band located between a slow and a fast bound. Since the relationship $U \simeq 0.7Lf$, given in Eq. (1), intrinsically relates the tail beat frequency to the swimming speed to a very good approximation, with a factor of 2 at most for fish and cetaceans, we can compare in the same figure the speed data with the prediction of the slow and fast bounds expected by our model. The model is based on Eqs. (1) and (4) together with the parameters used to fit the slow and fast bounds of the frequency measurements (Fig. 1). Although there are no free parameters for speed prediction, the match with biological data is good, highlighting the overall consistency of the approach and the model's predictive capacity in determining the minimum and maximum speeds achievable by natural swimmers.

Scaling laws are inferred for a given activity level:

$$U \simeq 0.7 f_0 L, \quad \text{if} \quad L \ll L_c \qquad (8)$$

$$U \simeq 0.7 f_0 L_c = 0.7 \sqrt{\sigma_0/\rho}, \quad \text{if} \quad L \gg L_c. \qquad (9)$$

For small animals, we therefore expect swimming speed to increase with length, whereas it should saturate at a constant value for large animals. This is consistent with studies that found a tail beat frequency scaling as $L^{-1}$ for large swimmers, and also found a nearly constant swimming speed in this case[18,49]. Still for large swimmers, the

swimming speed should range approximately between 1 and 7 m.s$^{-1}$ depending on the activity level, given the values of maximum frequency $f_0$ and crossover length $L_c$ found from the fits of the slow and fast bounds, respectively.

We compared our approach with the study by Hirt et al., which reported data on maximum swimming speed[50], to discuss our understanding of the burst activity level and the fast bound reached by swimmers. The two datasets differ in at least two aspects. First, ours contains only speed measurements that are associated to frequency measurements while Hirt et al. focused on speed measurements only. Second, the two datasets do not apply the same filters to select relevant data. Whereas our dataset makes no distinction as to activity level and is based solely on direct speed measurements provided by peer-reviewed studies, Hirt et al. only took into account references that dealt with the burst regime, without distinguishing whether the study was peer-reviewed or not, or whether the data were direct measurements or estimates. For $L \lesssim 0.5 - 1$ m, we observe that both datasets exhibit the same upper limit. For most lengths, the match is perfect and only small differences are observed, but they are at most a factor of 2. This regime is consistent with the scaling law expected in the limit $L \ll L_c$: quantitatively, Hirt et al. found that the maximum swimming speed scales as $U \propto M^{0.36}$, on average, or equivalently $U \propto L$, in agreement with the scaling law proposed in Eq. (8). For $L \gtrsim 0.5 - 1$ m, we observe a significant difference. While our scaling law inferred from frequency measurements predicts a constant maximum speed, around 5–10 m.s$^{-1}$, data gathered by Hirt et al. suggest a humped shape with a maximum around 30–40 m.s$^{-1}$ obtained for $L \simeq 1 - 3$ m followed by smaller speeds for larger animals. In fact, we propose that the two datasets differ in this two region for two different reasons.

1. While Hirt et al. did an enormous amount of work gathering data from the literature, we suggest that the maximum is artificial if we apply relevant filters. Most of the highest maximum speed data collected by Hirt et al., for fish ranging from about 1 to 3 m in length, are estimates or predictions

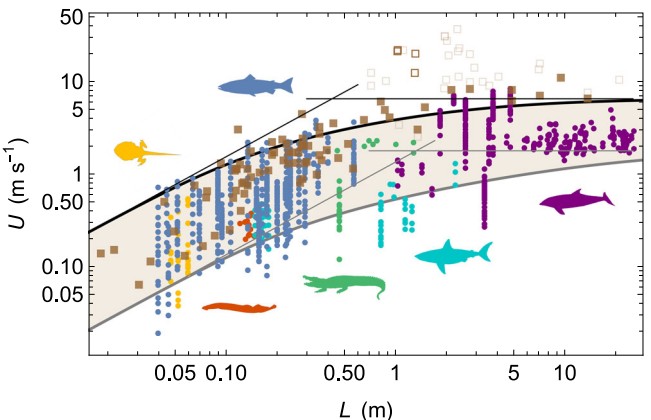

**Fig. 2 | Model predictions and observations of swimming velocity-length.** Swimming speed $U$ as a function of length $L$ (closed circles) for agnathans (red), cartilaginous fishes (cyan), ray-finned fishes (blue), lobe-finned fishes (pink), amphibians (yellow), reptiles (green), and mammals (purple). Brown squares correspond to the data gathered by Hirt et al. for maximum swimming speeds[50] using the mass-length relationship (see Methods). Open translucent squares represent either non-peer reviewed papers or data coming from estimations and not actual measurements. Open opaque squares represent data obtained using rod-mounted devices. The other data are represented by closed opaque squares. The black and gray thick lines represent the fast and slow bounds, respectively, as predicted by the model together with the parameters used to fit the frequency measurements in Fig. 1. Thin lines are the scaling laws in the limit of very small and very large swimmers. The alabaster area represents the speed band used by the swimmers.

based on in vitro physiological measurements. These measurements have been shown to significantly overestimate the expected maximum speed of what were thought to be the fastest swimmers, like *billfish*. First, these fish have lengths $L > L_c$ and consequently their tail beat frequencies are significantly smaller than the maximum frequency $f_0$ expected from the muscles: we predict that a 2 m-long swimmer with $f_0 = 20$ Hz would swim with a maximum tail beat frequency four times smaller, below 5 Hz (Fig. 1), and thus would have a maximum speed reduced by a factor of 4 in comparison to predictions based on the twitch contraction time only. Second, recent estimates based on measurements of twitch contraction times of anaerobic muscles also provide upper bounds lower than 10 m.s$^{-1}$ for *billfish* and other large marine predatory fish[14]. Actually, reported values of maximum speeds agree with this argument and refute some incredibly huge values that had been estimated for animals of this size. For *marlins*, direct measurements using speedometers showed that the observed maximum swimming speed was around 2.25 m.s$^{-1}$ in ref. 51, considerably lower than the estimates of 30 m.s$^{-1}$in ref. 52. Burst speeds of *sailfish* also show values around 8 m.s$^{-1}$ measured with high-speed video and accelerometry[28], a value much smaller than 30 m.s$^{-1}$in ref. 52. In addition, there is some theoretical evidence that the maximum speed should be smaller than 15 m.s$^{-1}$, because cavitation should appear at greater speeds, which should damage the flesh of the swimmer[53]. In Fig. 2, we have used open translucent squares to represent data that were not actual measurements but estimates, or that were taken from non-peer-reviewed studies (see Methods). If we remove these points from the analysis, we find that the two upper bounds of the datasets match very well, with the exception of four data points obtained for *tuna* and *barracuda* that are still significantly faster than our fit of the fast bound (open opaque squares in Fig. 2). Note that all of these points were measured using rod-mounted devices that measure the speed at which the line is pulled from the reel when a fish is hooked and pulling on the line (in refs. 5,54,55 and Methods). Given the large fluctuations in the measurements made with this method[54], it is likely that it overestimates the maximum speed, which is also supported by the fact that barracuda and tuna do not show particularly high maximum frequencies[14].

2. For $L \gtrsim 5$ m, if we remove from the Hirt et al. dataset data that are not estimates or have not been peer-reviewed, we observe that the remaining data of Hirt et al. exhibit systematically faster speeds than in our dataset (closed opaque squares in Fig. 2). We attribute this difference in a lack of measurements of the burst regime for $L \gtrsim 5$ m within our dataset. In the latter, we have retained only the speed measurements associated with the frequency measurements (see Methods). With this filter, the fastest speed around 4 m.s$^{-1}$ correspond to *baleen whales*[19,56], while swimming speeds up to 10 m.s$^{-1}$ were recorded in *sperm whales*[57], for which tail beat frequencies were unfortunately not measured, but which should be consistently higher than those plotted in Fig. 1 for the same length. Unlike *sperm whales*, *killer whales* and some other large marine predators, most cetaceans are filter feeders and do not have predators due to their size. Therefore, they do not often need to move at maximum speeds, which would favor data closer to the slow bound than the fast bound. This should explain why speed and frequency measurements for $L \gtrsim 5$ m in our dataset are systematically below the fast bound obtained from the fit over the whole range in length, while the re-filtered data of Hirt et al. (closed opaque squares in Fig. 2) give faster speeds, typically 5–10 m.s$^{-1}$, in

agreement with the fit of the fast bound obtained in our approach.

Following these considerations, it is reasonable to consider that the maximum speed is constant for size $L \gtrsim 0.5 - 1$ m, with a typical value around 5–10 m.s$^{-1}$, in very good agreement with the prediction $0.7 f_0 L_c$ inferred from the fit of the fast bound. From the definition of $L_c$, $U \simeq 0.7\sqrt{\sigma_0/\rho}$ (Eq. (9)) and the maximum swimming speed is directly associated with the maximum stress generated by the fast muscles to push the surrounding water. In their model, Hirt et al. state that heavier (and consequently longer) animals need more time to accelerate to achieve maximum speed and this fact would prevent the heaviest animals from being the fastest. Here we suggest that the effect of a finite acceleration time would be a second-order effect, unlike the other locomotion modes running and flying[50].

### Scaling the swimming power

The question of energy efficiency is more difficult to address due to the small amount of biological data in the literature that measures muscle power. Nevertheless, we can speculate on the main trends by following a few hypotheses. The power produced by the swimmer's metabolism must be at least of the same order as the power required to move the swimmer's body. An estimate of muscle power to produce the undulatory kinematics, $P \sim \rho L^5 f^3$, is obtained by multiplying the muscle force to push the fluid, Eq. (3), by the tail speed $Lf$, which approximates the speed of muscle contraction. From our study, we have shown that $f = f_0$ and $f = \sqrt{\sigma_0/\rho}L^{-1}$ for very small and large swimmers, respectively. This gives the following scaling laws in these two limits:

$$P \sim \rho f_0^3 L^5, \quad \text{if} \quad L \ll L_c \quad (10)$$

$$P \sim \rho^{-1/2}\sigma_0^{3/2}L^2, \quad \text{if} \quad L \gg L_c. \quad (11)$$

Muscle power is therefore an increasing function of length for a given level of activity, but the increase is significantly more pronounced for smaller swimmers.

Specific power $P_M = P/M$, defined as power per unit mass, provides information on how muscle fibres function for a given level of activity. From the allometric relation $M \sim \rho L^3$ (see Methods), we deduce $P_M \simeq L^2 f^3$. This estimate of the specific power is plotted in Fig. 3 as a function of length. Following the scaling laws of tail beat frequency, it increases as $f_0^3 L^2$ for small swimmers ($L \ll L_c$) and decreases as $\rho^{-3/2}\sigma_0^{3/2}L^{-1}$ for large swimmers ($L \gg L_c$). In these two limits, the

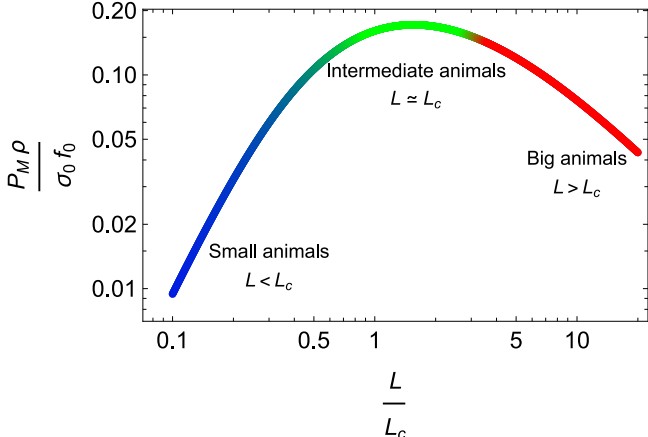

**Fig. 3 | Specific power-length graph.** Specific muscle power $P_M$ as a function of length, as estimated from the model. The curve is drawn with $\kappa = 1$ and is represented in its dimensionless form.

specific power produced by the muscle is negligible compared to that obtained for lengths around $L_c$. For $L \sim L_c$, the specific power is maximum with a value scaling as $\sigma_0 f_0/\rho$, which gives an estimate of the maximum specific power that can be delivered by the muscle. This graph highlights the fact that small and large swimmers do not use the full capacity of muscle power, unlike intermediate-sized swimmers.

Another quantity of interest is the cost of transport (COT), which measures the energy spent per unit mass and distance traveled. It writes $COT = P/(UM)$, which can be estimated as $Lf^2$. Our approach suggests that the COT exhibits a maximum around $L_c$ as well, with two limits that can be expressed as:

$$COT \sim Lf_0^2, \quad \text{if} \quad L \ll L_c \tag{12}$$

$$COT \sim \frac{\sigma_0}{\rho}\frac{1}{L}, \quad \text{if} \quad L \gg L_c. \tag{13}$$

In the limit of large swimmers, $L \gg L_c$, Eq. (13) retrieves the prediction proposed in[18], i.e. $COT \sim M^{-1/3}$, in agreement with the few measurements available in the literature[58].

## Discussion

Based on our results, we can conclude that swimming frequency and speed in natural swimmers are primarily determined by their length and activity level. We have developed a simple model that takes into account biological characteristics, such as muscle fiber type and Hill's muscle model, as well as the interaction of the undulating swimmer with its environment, to explain the collected data. This model only requires a few parameters, which can be further refined in the future to incorporate specific characteristics of each swimmer, such as body temperature and swimming gait.

Our study highlights a crossover at a length of approximately $L_c \sim 0.5 - 1$ m, which separates two distinct limits. Small swimmers with length $L \ll L_c$ are primarily constrained by biological factors. Conversely, large swimmers are constrained not only by biology but also by their surrounding environment. For a given activity level, different scaling laws are observed for swimming frequency and speed in these two limits. For small animals with length $L \ll L_c$, the tail beat frequency is limited by the fastest frequency $f_0$ that the muscle fibers can reach during contraction. On the other hand, for large animals with $L \gg L_c$, the muscle cannot operate at its maximum frequency but rather works at its maximum force to overcome the resistance of the surrounding water. The muscular force scales as $\sigma_0 L^2$, while the thrust scales as $\rho L^4 f^2$, leading to a tail beat frequency $f$ that decreases with length following the scaling $f \sim L^{-1}$. While these considerations determine the tail beat frequency, swimming speed is predicted through the relationship $U = 0.7Lf$.

This distinct behavior also extends to other quantities, such as muscle power for locomotion. According to our model, very small and large swimmers exhibit scaling laws of $L^5$ and $L^2$ for muscle power, respectively. To test these predictions, measurements of oxygen consumption over a wide range of lengths and activity levels[24,25] could be conducted. Additionally, in the framework of our model, $L_c$ also marks a significant change in the way muscles are utilized. Small swimmers utilize muscles at their maximum speed but with negligible force compared to their maximal capabilities, while large swimmers exhibit the opposite behavior. Very small and large swimmers do not utilize their full power capabilities compared to the maximum power available. Interestingly, intermediate-sized fish with lengths around $L_c$ would need to utilize their muscles at their full capacity to undulate and move efficiently through water. Moreover, these intermediate-sized swimmers are predominantly heterotherms, which may indicate a fine-tuning of their muscles that already work at maximal power. In light of this observation, it is worth exploring in future studies whether intermediate fish are more likely to employ economical locomotion

strategies (e.g., intermittent swimming, schooling, etc.) compared to very small and large swimmers.

In conclusion, our study reveals that swimming frequency and speed in natural swimmers are primarily determined by length and activity level. By developing a simple model incorporating biological characteristics and environmental interactions, we were able to explain the observed data. The distinct behaviors observed in small and large swimmers, as well as the transition at a critical length, shed light on the intricate dynamics of swimming. These findings not only advance our understanding of animal locomotion but also provide valuable insights for the design of biomimetic and autonomous swimming robots[59–63].

## Methods

We have compiled a comprehensive glossary in Table 1, which succinctly presents all the definitions and notations utilized throughout this article.

### Data and allometry plots

We compiled a comprehensive length-frequency database consisting of 1216 animals from various species, morphologies, and sizes. To construct this dataset, we gathered data from reviewed articles that measured length and frequency. Among the references, we also recovered swimming speed data when these were available. We organized the data based on the division of vertebrates presented in the phylogenetic tree shown in Fig. 4a. The divisions include agnathans, cartilaginous fishes, ray-finned fishes, lobe-finned fishes, amphibians, reptiles, and mammals. In cases where length data were not explicitly reported but the mass of the animals was available[16], we calculated the length using the allometric relation derived in Fig. 4b: $L = 0.44M^{0.33}$ (where $L$ is in meters and $M$ is in kilograms), assuming geometric similarity. Additionally, we obtained length-amplitude and length-mass data. The references for the data sources are provided in Table 2.

Our approach is based on three relationships that we have verified using the data presented in Tab. 2.
1. Geometric similarity for aquatic animals: Fig. 4b depicts the geometric similarity for 416 different individuals. The best fit of the

### Table 1 | Glossary

| Notation | Quantity |
|---|---|
| $\lambda$ | Deformation wavelength |
| $A$ | Tail beat amplitude |
| $f$ | Oscillating frequency |
| $L$ | Length of the animal |
| $U$ | Swimming speed |
| $F$ | Force delivered by the muscle |
| $F_0$ | Maximal force produced by the muscle in the Hill's model |
| $v$ | Muscle contraction velocity |
| $v_0$ | Maximal muscle contraction velocity in the Hill's model |
| $\kappa$ | Parameter of the Hill's model |
| $\sigma_0$ | Typical stress produced by the muscle |
| $f_0$ | Maximal oscillating frequency of the muscle in the Hill's model |
| $\rho$ | Density of the water |
| $L_c$ | Critical length |
| $c$ | Body wave speed. |
| $R$ | Radius of the elastic beam |
| $f_e$ | Natural frequency of bending waves |
| $M$ | Mass of the animal |
| $P$ | Power produced by the muscle in the Hill's model |
| $P_M$ | Specific power: power per unit mass |
| COT | Cost Of Transport |

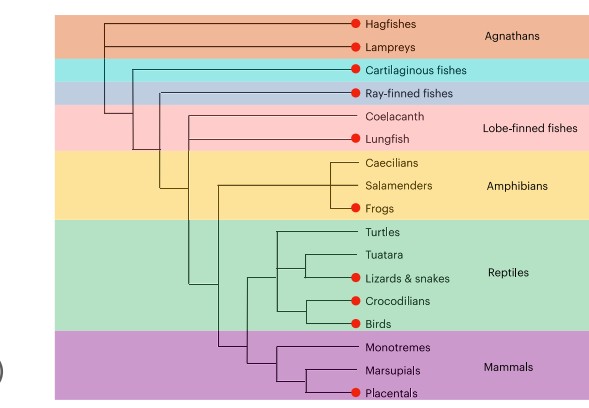

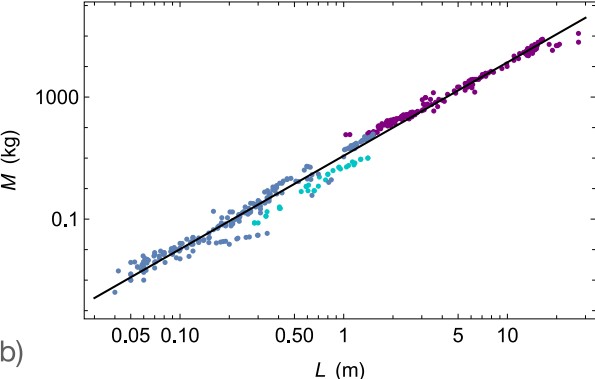

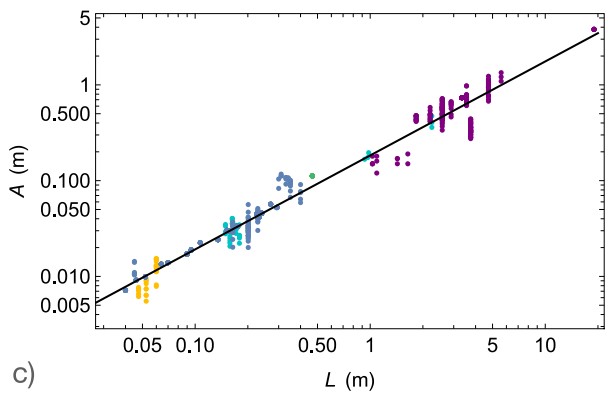

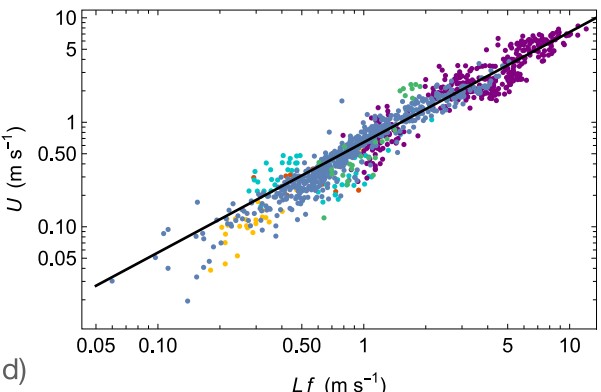

**Fig. 4 | Allometric plots. a** Representation of the swimmers used in the dataset in the vertebrate Phylogenetic tree. The red disks correspond to species whose swimming kinematics have been measured and used in our dataset. **b** Animal mass and **c** tail beat

amplitude as functions of animal length. The solid lines represent the best power-law fits of the data. **d** Swimming speed as a function of the product of length and tail beat frequency. The solid line represents the best power-law fitting the data.

data yields $M = (11.92 \pm 0.35)L^{3.06 \pm 0.02}$, where $M$ is measured in kilograms and $L$ in meters ($R^2 = 0.99$). This result is consistent with the geometric similarity characterized by an exponent of 3, which extends Economos's relation ($M = 11.27L^{2.9564}$) over four orders of magnitude in length or ten in mass. By enforcing the exponent to be exactly 3, we obtain $M = (11.84 \pm 0.35)L^3$ or its dimensionally homogeneous form $M = (0.0119 \pm 0.0003)\rho L^3$ with $\rho = 1000$ kg.m$^{-3}$.

2. Proportionality between $A$ and $L$ : in Fig. 4c, we plotted $A$ as a function of $L$ and verified that $A$ is proportional to $L$, which was initially demonstrated by Bainbridge[5]. Our study extends this

relationship to 365 different specimens across four orders of magnitude in length. The best fit with a power law yields $A = (0.185 \pm 0.009)L^{0.981 \pm 0.005}$, indicating that both quantities are proportional ($R^2 = 0.98$). The best proportionality relation is given by $A = (0.188 \pm 0.001)L$.

3. Relationship between $U$ and $Lf$: we plotted $U$ as a function of $Lf$ in Fig. 4d, utilizing a database consisting of 1086 individuals. The best fit with a power law yields $U = (0.64 \pm 0.02)(Lf)^{1.06 \pm 0.01}$, with an $R^2$ value of 0.95. Assuming an exponent of 1, the fit gives $U = (0.706 \pm 0.005)Lf$. In cases where length data were unavailable but mass was known, we converted the data using the relationship derived in Fig. 4b. The exclusion of data with an unknown length did not have a significant impact on the fit parameters.

**Table 2 | References reporting relations between length, frequency, amplitude or mass of swimmers**

| | Frequency vs length or mass | Speed | Amplitude vs length | Mass vs length |
|---|---|---|---|---|
| Agnathans | 67, 68 | 68 | - | - |
| Cartilaginous fishes | 17, 69–71 | 69–71 | 69, 70 | 72 |
| Ray-finned fishes | 5, 7, 17, 26 | 5, 7, 26 | 5, 7, 26, 73, 74 | 5, 75–78 |
| | 28, 73, 74, 79–81 | 73, 74, 79 | 73, 74, 79 | |
| Lobe-finned fishes | 82 | - | - | - |
| Amphibians | 83 | 83 | 83 | - |
| Reptiles | 49, 84, 85 | 49, 84 | 49, 84 | - |
| Mammals | 6, 16, 27 | 6, 27 | 6, 27, 86, 87 | 27, 86–90 |
| | 56, 86, 87, 91 | 56, 86, 87, 91 | | |

**Hill's muscle model**

Hill's equations for tetanized muscle contraction (Eq. (2)) can be rewritten to express the force in the muscle $F$ as a function of the swimming frequency $f$:

$$\frac{F}{F_0} = \frac{1 - \frac{f}{f_0}}{1 + \kappa \frac{f}{f_0}}. \tag{14}$$

$F_0$ is the maximum isometric force generated in the muscle and $f_0$ is the maximum tail beat frequency. In its dimensionless form, the force $F/F_0$ is a decreasing and convex function of the frequency $f/f_0$, whose degree of curvature is quantified by the parameter $\kappa$ (Fig. 5). From our analysis, very small animals ($L \ll L_c$) swim at maximum frequency and negligible force ($f = f_0$ and $F \ll F_0$), while very large animals ($L \gg L_c$)

swim at negligible frequency and maximum force ($f \ll f_0$ and $F = F_0$). Intermediate-sized animals ($L \sim L_c$) swim at intermediate frequency and force ($f \lesssim f_0$ and $F \lesssim F_0$).

## Characterization of the burst and sustained activity levels

In this section, we will elaborate on how we determined the fast curve $[f]_{\text{fast}}(L)$ and the slow curve $[f]_{\text{slow}}(L)$ that encompass the band in the $L - f$ plane. The process of determining these bounds is not straightforward due to the presence of various orders of magnitude, which necessitates the use of logarithmic scales. To begin, we divided the $L$ axis into $N$ intervals of equal size on a logarithmic scale. Within each interval, we identified the minimum $f_{\text{min}}$ and maximum $f_{\text{max}}$ frequencies. Here is how we determined each bound:

- Fast bound $[f]_{\text{fast}}(L)$: We calculated the average of all data points within each interval with frequencies $f$ such that $0.9 f_{\text{max}} < f < f_{\text{max}}$.
- Slow bound $[f]_{\text{slow}}(L)$: Similarly, we averaged all the data points within each interval with frequencies $f$ such that $f_{\text{min}} < f < 1.1 f_{\text{min}}$.

By following this approach, we obtain a dataset that approximates the upper curve $[f]_{\text{fast}}(L)$ as well as the lower curve $[f]_{\text{slow}}(L)$. To estimate the three parameters of the model ($L_c$, $f_0$, and $\kappa$) for each bound, we fit the $N$ averaged points using Eq. (4) and employed the least absolute deviations (LAD) method[65]. This method is less sensitive to

outliers compared to the standard least squares method. To ensure robustness, we varied the value of $N$ from 10 to 50. We then averaged the best parameter values ($L_c$, $f_0$, and $\kappa$) and estimated the standard deviation to characterize each parameter. This range in $N$ allows for a statistically significant number of points while avoiding empty intervals.

Finally, we checked that the full model given by Eq. (4) provides best results with respect to the two limits $f \propto 1/L$ and $f = \text{cst}$. Quantitatively, we computed the mean AIC values ($N$ varying from 10 to 50) and the corresponding error for the three cases[66]. We summarize these computations in Table 3, and we note that our model gives the best results, both in AIC values and absolute errors.

## Representation of the swimmers used in the dataset in the vertebrate phylogenetic tree

We gathered data for almost all the clades as reported in the phylogenetic tree shown in Fig. 4a.

## Water temperature and thermoregulation

We represent in Fig. 6a and b the $L - f$ measurements of Fig. 1 with the additional information of thermoregulation properties and water temperature when provided.

## Filtering of maximum speed data

We investigated the origin of the data gathered by Hirt et al.[50] to establish objective criteria on data selection. Of all the measures reported in the study, we found three classes of data that were not as reliable as the others. First, we identified data coming from non-peer reviewed papers. Second, some of the data are only estimates, not actual measurements. Third, we found that all data obtained with rod-mounted devices are above the main trend, which could be artificial due to the high fluctuations in these measurements[54]. All these three classes of data points are summarized in Table 4.

## Reporting summary

Further information on research design is available in the Nature Portfolio Reporting Summary linked to this article.

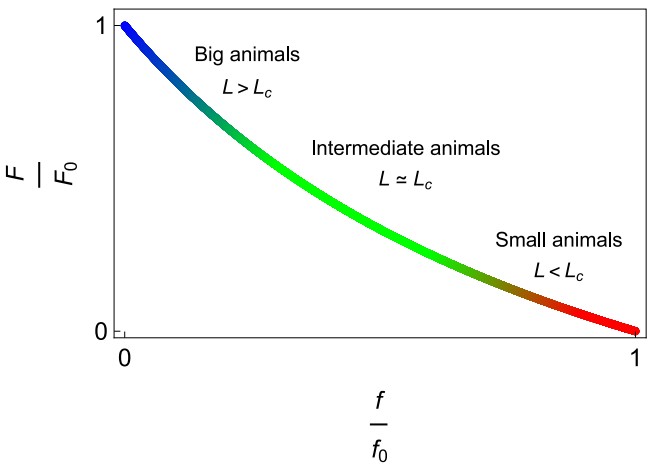

**Fig. 5 | Hill's muscle model.** Muscle force as a function of tail beat frequency. The curve is drawn with $\kappa = 1$ and is represented in its dimensionless form.

**Table 3 | Summary of the average AIC and Error for the fitting of the boundaries with three different laws**

|  | AIC | | | Error | | |
|---|---|---|---|---|---|---|
|  | **Model** | **$f \sim 1/L$** | **$f = \text{cst}$** | **Model** | **$f \sim 1/L$** | **$f = \text{cst}$** |
| fast bound | 146 | 171 | 205 | 1.9 | 4.0 | 5.80 |
| slow bound | 26 | 61 | 81 | 0.24 | 0.61 | 0.67 |

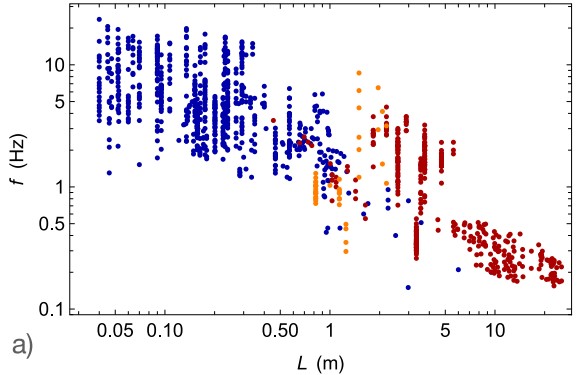

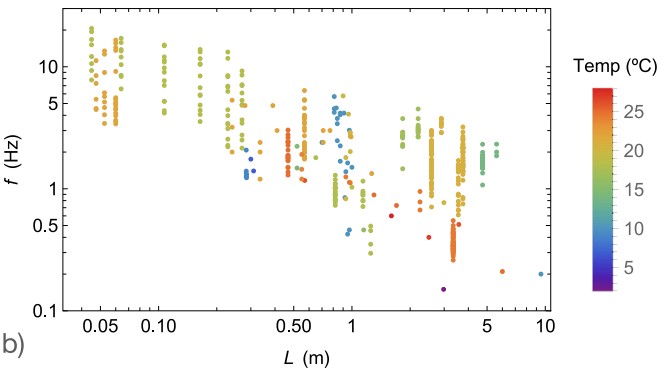

**Fig. 6 | Water temperature and thermoregulation. a** Tail beat frequency $f$ as a function of length $L$ with coloring indicating the type of thermoregulation. The ectothermics, endothermics and heterothermics animals are shown in blue, red and orange, respectively. **b** Tail beat frequency $f$ as a function of length $L$ with coloring indicating water temperature.

**Table 4 | Data used in Hirt et al.[50] that are either provided by non-peer-reviewed (NPR) articles, given as estimates or obtained using rod-mounted devices**

| Animal | Mass (kg) | Length (m) | Maximum speed (m.s⁻¹) | Speed reference | Dismission reason |
|---|---|---|---|---|---|
| *Aptenodytes patagonicus* | 14 | 1.05 | 3.36 | 92 | NPR |
| *Pygoscelis antarcticus* | 4.5 | 0.72 | 8.89 | 92 | NPR |
| *Pygoscelis papua* | 8.2 | 0.88 | 10 | 92 | NPR |
| *Acanthocybium solandri* | 13.31 | 1.03 | 21.4 | 55 | Rod and reel |
| *Acanthocybium solandri* | 16.64 | 1.11 | 21.39 | 92 | NPR |
| *Carcharodon carcharias* | 800 | 4.04 | 11.11 | 92 | NPR |
| *Galeocerdo cuvier* | 550 | 3.57 | 8.89 | 92 | NPR |
| *Istiompax indica* | 150 | 2.32 | 36.11 | 51 | Estimation |
| *Istiophorus albicans* | 90 | 1.95 | 30.56 | 92 | NPR |
| *Istiophorus albicans* | 90 | 1.95 | 30 | 51 | Estimation |
| *Isurus oxyrinchus* | 105 | 2.06 | 18.8 | 93 | NPR |
| *Isurus oxyrinchus* | 300 | 2.92 | 13.89 | 92 | NPR |
| *Makaira nigricans* | 153.5 | 2.33 | 20.83 | 51 | Estimation |
| *Sphyranea argentea* | 4.5 | 0.72 | 12.22 | 92 | NPR |
| *Sphyranea barracuda* | 26.56 | 1.29 | 12.19 | 5 | Rod and reel |
| *Tetrapturus audax* | 163 | 2.38 | 22.5 | 92 | NPR |
| *Thunnus albacares* | 13.11 | 1.03 | 20.83 | 54 | Rod and reel |
| *Thunnus orientalis* | 250 | 2.74 | 19.44 | 92 | NPR |
| *Thunnus thynnus* | 27.22 | 1.31 | 19.67 | 5 | Rod and reel |
| *Xiphias gladius* | 98 | 2.01 | 26.94 | 92 | NPR |
| *Balaenoptera musculus* | 108400 | 20.75 | 10.30 | 31 | Estimation |
| *Delphinus delphis* | 95.32 | 1.99 | 10.30 | 31 | Estimation |
| *Enhydra lutris* | 30 | 1.35 | 2.5 | 92 | NPR |
| *Megaptera novaeanglia* | 30000 | 13.53 | 7.5 | 92 | NPR |
| *Orcinus orca* | 4100 | 6.97 | 13.33 | 92 | NPR |
| *Orcinus orca* | 4300 | 7.09 | 15.4 | 94 | NPR |
| *Pusa hispida* | 88.07 | 1.94 | 8.33 | 95 | NPR |
| *Tursiops truncatus* | 250 | 2.74 | 9.72 | 92 | NPR |
| *Zalophus californianus* | 158 | 2.36 | 11.11 | 96 | NPR |
| *Dermochelys coriacea* | 420 | 3.26 | 9.8 | 94 | NPR |

We list the type of animal together with the corresponding mass and speed, the length using the mass-length relationship if not provided, and references.

## Data availability

The data used in this study are available in the Supplementary Data 1, and Source Data files, or from the corresponding author upon request. Source Data for Figs., 1, 2, 4b, c and d, 6a and b are available as a Source Data file. Source data are provided with this paper.

## Code availability

The code for computing the external bounds is available in the Supplementary Code 1 file.

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

## Acknowledgements

We are grateful to François Gallaire, Guillaume Allibert and José Luis Trejo for enlightening discussions. JSR acknowledges funding from Ministerio de Universidades and European Union-NextGenerationEU. This work was supported by the French government, through the UCAJEDI Investments in the Future project of the National Research Agency (ANR-15-IDEX-01).

## Author contributions

J.S.R. gathered all the data corresponding to biological measurements. J.S.R., C.R. and M.A. conceived and developed the theoretical model. J.S.R., C.R. and M.A. wrote the manuscript. C.R. and M.A. supervised the project.

## Competing interests

The authors declare no competing interests.
