## [Peer Review File · Nature Communications]

Scaling the tail beat frequency and swimming speed in underwater undulatory swimmingReviewers' Comments:

Reviewer #1:

Remarks to the Author:

See attached.

Comments for authors

This is an interesting manuscript that attempts to develop a scaling relationship for tail beat frequency across a wide range of sizes of swimming animals. The authors suggest that there are two different regimes for small and large animals, in which frequency is independent of size for small animals, but decreases with increasing size for large animals. This leads to the suggestion that only intermediate size animals are able to maximize muscle power at high swimming speeds.

I have two major criticisms.

1. I think the authors have done a good analysis in the end, but I found the manuscript very difficult to follow. I think I would find it clearer if the section on p 6 that explains the two limiting frequency domains came first, and then the overall relationship as given in Eq 2-5 came next. I've tried to highlight points where I was confused in the minor comments below.
2. I do not understand the explanation of c speed in the $L \gg L_c$ bound (p. 6, 4th paragraph, lns 1-10). Why does it matter that $L_c f_0$ has units of speed? If this is meant to be the body wave speed, then that should be explained more clearly. The second explanation in the paragraph, about the scaling of a natural bending frequency, makes more sense to me. Why give multiple explanations for this scaling relationship?

Minor comments

1. Overall – please number lines in your manuscript to make it easier for reviewers to give comments.
2. P. 1. $U \cong 3.3Af$. This is not a standard way of writing the swimming speed, although it may be true. The expression later on the page ($U \cong 0.7Lf$) is much more typical, and would be the better place to start.
3. P. 1-2, last paragraph. Is the point here just that many factors affect swimming speed and tail beat frequency? I found this paragraph confusing.
4. P. 5 and throughout. I think it would be clearer to use F_{max} rather than F_0 , and v_{max} and f_{max} similarly.
5. P. 5, "...expect the scaling $v \sim Af$ ". This is OK, but it depends on the fact that A does not vary (much) with f . There may be some variation in A as a function of swimming speed; see the supplemental information in di Santo et al. (Di Santo et al., 2021), for example.
6. Eq 4 and 5. I was initially confused by Eq 4 and 5, because it is not immediately obvious that f cannot be larger than f_0 (in the current notation). Using f_{max} would make this clearer.
7. P. 6, end of 4th paragraph. Refs 35-37 are all mathematical or physical models and suggest that flexible panels or swimmers with homogeneous mechanical structures may have resonant bending modes, but they do not actually show that animals, whose bodies are much more complex, have resonant modes or that they match them while swimming.
8. P. 11, end of 1st full paragraph. If this model is true, it suggests that intermediate size fish may be the only ones able to take advantage of the full range of muscle power. This is a really important point, and probably deserves more emphasis. Given existing measurements of cost of transport, is it possible to test this idea against data?
9. P. 11, first paragraph in section 4.1. The scaling relationship $L = 0.44M^{0.33}$ has a lot of scatter. Is it possible to test the model only against data in which the length is known?
10. Section 4.1. Please statistically test the scaling relationships against known allometric patterns (eg. L should scale as $M^{\frac{1}{3}}$ and A as $L^{1.0}$).

11. Fig. 4. This is an important implication of the model, and I think it should be somewhere other than in the methods.
12. P. 14. What is the LAD method? Please give a citation.
13. Reference 4 is by di Santo, V. not Santo, V.D.

Reviewer #2:

Remarks to the Author:

Please see attached pdf of my review

March 5th 2023

Objet: review of the manuscript *Scaling the tail beat frequency and swimming speed in underwater undulatory swimming* (NCOMMS-23-03609), submitted to Nature Communications.

It was a pleasure reviewing this meta-analysis of vertebrate undulating frequencies and speeds. Overall, I found the paper well-written, with regard to both language and style, and the results compelling. I was a little disappointed, however, to not find more information in the method section, as well as the data and analysis code available. I have outlined below some of the strengths and weaknesses of the manuscript, as well as several concerns/comments. Since the pdf file has no line numbering, I also include my annotated copy, showing a few specific minor comments.

Strengths

- The study is based on a large number of data points gathered from the literature on different vertebrates using undulatory swimming motions: amphibians, reptiles, fish, birds and mammals. The authors have included all the observations despite variability in activity level and environmental conditions to fully considered the dispersion in the relationship between undulatory frequency and body length.
- The authors aim at establishing a scaling law between oscillation frequency and body length which can apply to all vertebrates exhibiting undulatory swimming motion, considering the mechanical and biological properties of the muscles and the surrounding fluid. In my opinion, this makes for a novel and comprehensive approach. I am not an expert on muscle physiology, but their rationale and interpretation of Hill's muscle model seems sound.

Aspects to improve

- I recommend including a glossary for the main parameters used. Even if those are defined in the equations, they are used a lot within the main text and it can become daunting to search through the text/equations for their meaning. Alternatively, authors may consider writing in full the parameter name (i.e: frequency instead of f_0) more often. It will make the text easier to follow.
- How was the length cross-over determined? From my reading, it seems it was determined visually from the scatterplot. If you have a change of slope in your relationship between frequency and length, I would prefer to see a more formal way to identify the breakpoint, for example by applying a moving-point regression to the relationship between frequency and length, with the breakpoint as the 'moving' variable in the model (fitting a model for each possible value for the breakpoint). A plot of AIC values for each of the fitted models will give you the location (likely an area of a few possible locations) for the change of slope.

- Some discussion is needed on the nature of the data used in the modelling effort, and how it can impact the results.
- I feel the discussion ends abruptly. A few concluding sentences could be useful to wrap things up.
- What are the limitations of your model (s), if any?
- Method section is hard to follow, especially the definition of the fast and slow bounds.

Specific concerns / comments

- I would appreciate seeing a table (likely in supplemental material) listing the vertebrate classification, species, wild/reared, sample size, the experimental apparatus, range of body lengths, temperatures, speed & frequencies observed, and the references for the data gathered from the literature. As it is, the manuscript offers little information on the nature of the data used in the modelling. I assume most data comes from videography. The authors are making the efforts of filtering out some data that are considered likely biased, or simply not precise enough. However, it would be interesting to know better what kind of data were retained. Are we talking of mostly individuals swimming in small flow tanks or respirometers? Open flumes? Directly in the field? Did you mostly collect frequencies data from fish over 1-2 tailbeats, or was the frequency averaged over a longer swimming duration?
- There are multiple empirical evidence pointing towards the fact that fish swim more efficiently when they can do so volitionally in an unconfined space, for example in open channels (or a river). Volition and an unconfined space likely affect their behavior, as well as their endurance and actual performance (swim speed achieved, distance swam). Some studies have found migratory diadromous species optimizing their swim speeds when facing velocity challenges (i.e going up a river or long pelagic migrations), which may suggest some optimization mechanism for undulation frequency. Given this, it is more than likely that they may modulate their undulation frequency in an adaptive response to the environment. What does it mean regarding your results? Given the data retained, could you be underestimating or overestimating the slow and fast bounds?
- The swim speeds shown in Figure 2 are estimates based on the law $U = 0.7 Lf$ and some actual data from Hirt et al. are shown. There was no swim speed data available in the literature, along with the length and frequencies data gathered by the authors? If so, it would be interesting to also see how those observed swim speeds fit with the estimates from the model (s).
- How do you explain the stacking of points for some specific lengths, especially visible in amphibians and fish? Many individuals of the exact same length, or repeated observations from some individuals? If these represent repeated observations from the same individuals, was that

considered in the models, for example by adding a random component? Change in frequency for individuals of the same length can certainly mean a change in activity level (increase or decrease of swim speed), but also individual variability in some cases. Data points from mammals and birds are more evenly distributed.

- Data likely came from organisms swimming in a range of water temperatures, some endotherms, some ectotherms. Given body temperature will affect the muscles and water temperature the fluid properties, I was expecting more discussion, maybe even a figure, involving temperature. I feel it would help the interpretation of the data.
- Fig 1: I would like to see some discussion about the data points that lie outside of the upper and lower bounds. Several of them are from fish species. Do they represent some species that have distinct morphological characteristics, or individuals swimming under specific temperature conditions?
- In the discussion and method section 4.2, you mention the difference between how small, intermediate, and large animals use their muscles, and how it influences delivered muscle power for locomotion. Can you develop more this topic, perhaps in the discussion? Why do small individuals use negligible force and maximum frequencies, and large ones the opposite? Is this behavior constrained biomechanically? What are the implications of maximizing power for intermediate individuals? I feel more detail here will help understand better the relationship between tail-beat frequency and length and explain the location of the length cross-over.
- Where can we find the data and code used for this study? Will they be available on a repository? It is not clear to me.

Sincerely,

Elsa Goerig, PhD

Museum of Comparative Zoology
Harvard University
26 Oxford St.
Cambridge (MA) 02138
USA

Reviewer #3:

Remarks to the Author:

Sanchez-Rodriguez et al. provide an extensive review of swimming speed frequencies from the literature that they employ to assess the scaling of the tail beat frequency and swimming speed in aquatic vertebrates. The authors claim to propose a holistic model which might explain the diversity of scaling relationships proposed in previous studies. While I think the topic is of high relevance, I feel the authors might better sustain their conclusions by considering the points I detail below before the manuscript is considered for publication.

1. Data availability. In my opinion, the authors should make available all the raw data (not only the references of the works containing such data) and results. This is a good transparency practice, first, to facilitate repeatability and, secondly, to enable the revision process. Because of this, I could not check some of the following concerns.

2. Alternative models. I wonder whether the authors checked the fit of alternative models and how they were compared between them (AIC?). This is a critical point as it could be the case that a simpler model (with one slope) fits the data better than one considering a crossover point (with two slopes). Qualitatively, in Fig. 1 the change in slope claimed by the authors is not very evident. How did the authors determine the crossover values?

3. Data independence. Any analysis considering biological species fail in the statistical assumption of independency given that some species are more closely related between them than with the others, and therefore the points represented in Fig. 1 and Fig. 2 cannot be treated as independent data. Therefore, I highly encourage the authors to perform some additional analyses to control for this potential bias. I would recommend them to implement phylogenetic independent contrasts to discard the possibility that the pattern recovered is not just an artifact of phylogenetic structure.

4. Other contributing factors. I wonder why the authors did not include in the model other factors that they acknowledge in the introduction and discussion as potential key contributors of the tail beat frequency. The most obvious for me, given the point distribution in Fig. 1, is body temperature. Most taxa above the proposed threshold of 0.5-1 m are endothermic animals (birds and cetaceans). How can the authors discard the possibility that differences tail beat frequency are not just due to differences in thermophysiology and uneven representation of sizes for ecto- and endotherms?

5. Finally, I supposed, because of the animal outlines in the Fig. 1, that authors included aquatic birds (penguins) in their analyses. However, as far as I know penguins swim mostly by using their forelimbs, not by beating the tail.

1. Reviewer #1

RC: *This is an interesting manuscript that attempts to develop a scaling relationship for tail beat frequency across a wide range of sizes of swimming animals. The authors suggest that there are two different regimes for small and large animals, in which frequency is independent of size for small animals, but decreases with increasing size for large animals. This leads to the suggestion that only intermediate size animals are able to maximize muscle power at high swimming speeds.*

AR: The authors thank the reviewer for this positive report.

RC: I have two major criticisms

- I think the authors have done a good analysis in the end, but I found the manuscript very difficult to follow. I think I would find it clearer if the section on p 6 that explains the two limiting frequency domains came first, and then the overall relationship as given in Eq 2-5 came next. I've tried to highlight points where I was confused in the minor comments below.

AR: We agree that the result presentation was difficult to access in the submitted version of the manuscript. In the revised version, we have moved the paragraph (from section 2.3) that depicts the two boundaries in frequency into the section 2.1, i.e. before the Eq. 2-5, in order to better guide the reader in grasping the influence of the two kind of muscles. The detailed explanation remains in the section 2.3.

- I do not understand the explanation of c speed in the $L \gg L_c$ bound (p. 6, 4th paragraph, lns 1-10). Why does it matter that $L_c f_0$ has units of speed? If this is meant to be the body wave speed, then that should be explained more clearly. The second explanation in the paragraph, about the scaling of a natural bending frequency, makes more sense to me. Why give multiple explanations for this scaling relationship?

AR: Indeed, the scaling is equivalent to the scaling of the bending frequency. In the new manuscript, we explicitly state it. We have changed this paragraph, see lines 270 to 285:

For $L \gg L_c$, the frequency decreases with length as a consequence of the interplay between another biological constraint, here σ_0 , and the interaction of the swimmer with its environment [16, 18, 31]. In this limit, the model predicts that the frequency is given by $f = cL^{-1}$, where the parameter $c = L_c f_0 = \sqrt{\sigma_0/\rho}$ represents the body deformation speed. We estimate $[c]_{\text{slow}} \simeq 2.0 \text{ m}\cdot\text{s}^{-1}$ and $[c]_{\text{fast}} \simeq 9 \text{ m}\cdot\text{s}^{-1}$ for the slow and fast bounds, respectively. This scaling is obtained by approximating the swimmer by an elastic beam of length L , radius R and Young Modulus E . In such a case, the natural frequency of bending waves scales as $f_e \sim \frac{R}{L^2} \sqrt{\frac{E}{\rho}}$ [36]. By assuming geometrical similarity, $R \sim L$, and body elasticity compatible with stresses generated by muscle fibers $E \sim \sigma_0$, the natural bending frequency scales as the tail beat frequency f of large animals, defined in Eq. (6). This suggests that large animals undulate their bodies to resonate with their natural bending modes, as evoked by various studies [37 – 39], even though, this hypothesis have not been checked for biological swimmers.

RC: *Minor comments*

1. Overall – please number lines in your manuscript to make it easier for reviewers to give comments.

AR: We have numbered the lines of the manuscript to help the reviewers

2. P. 1. $U = 3.3Af$. This is not a standard way of writing the swimming speed, although it may be true. The expression later on the page ($U = 0.7Lf$) is much more typical, and would be the better place to start.

AR: We thank the referee for this comment. We wrongly cited Bainbridge in the $U \simeq 3.3Af$ equation since he did not write explicitly this relation in his work. We would rather use the guiding principle developed since Triantafyllou based on the constancy of the Strouhal number $St = Af/U$ for fish and cetaceans, from which the relation $U \propto Lf$ is derived as a consequence. Consequently, we have got rid of the wrong reference and highlighted the $U = 0.7Lf$ equation due to its widespread use.

3. P. 1-2, last paragraph. Is the point here just that many factors affect swimming speed and tail beat frequency? I found this paragraph confusing.

AR: Indeed, we have added a sentence at the beginning of the paragraph to explicitly state it (see line 101):

Second, many factors might influence the exact frequency range such as the previous training or the condition of the animal, its age or sex, and the water or body temperature [20–22].

4. P. 5 and throughout. I think it would be clearer to use F_{max} rather than F_0 , and v_{max} and f_{max} similarly.

AR: This remark is also pertinent. Initially, we used this notation, but, we preferred, for sake of clarity to use the standard notation with the o , as introduced in the Hill's model, because we reserved the wording "max" for max frequency of the upper bound of the interval. We respectfully keep the actual notation in order to avoid any confusion with what is the maximal frequency. This is also the standard notation in the original Hill's article.

5. P. 5, "...expect the scaling $v = Af$. This is OK, but it depends on the fact that A does not vary (much) with f . There may be some variation in A as a function of swimming speed; see the supplemental information in di Santo et al. (Di Santo et al., 2021), for example.

AR: We do agree that the amplitude slightly depends on frequency. We have added the following sentence at line 208:

Since the amplitude does not correlate significantly with quantities such as the swimming speed and can be considered constant for a given swimmer [4], this leads to $v/v_0 = f/f_0$ with f_0 the maximum tail beat frequency expected from the physiological limit of the muscle.

6. Eq 4 and 5. I was initially confused by Eq 4 and 5, because it is not immediately obvious that f cannot be larger than f_0 (in the current notation). Using f_{max} would make this clearer.

AR: see our answer on point 4.

7. P. 6, end of 4th paragraph. Refs 35-37 are all mathematical or physical models and suggest that flexible panels or swimmers with homogeneous mechanical structures may have resonant bending modes, but they do not actually show that animals, whose bodies are much more complex, have resonant modes or that they match them while swimming.

AR: We agree with the reviewer point of view: this is the reason why we use the verb suggest, because there is no formal demonstration of this resonance mechanism for real animals. In consequence, we have added the following text in the paragraph:

even though, this hypothesis have not been checked for biological swimmers.

8. P. 11, end of 1st full paragraph. If this model is true, it suggests that intermediate size fish may be the only ones able to take advantage of the full range of muscle power. This is a really important point, and probably deserves more emphasis. Given existing measurements of cost of transport, is it possible to test this idea against data?

AR: We have added a new section (line 492) that discusses the muscle power and the cost of transport (COT). There is not a large number of experiments that measures the COT. Our model predicts quantitatively the scaling for the COT to be proportional to the inverse of the length of the individual for $L > L_c$. We have added a relevant reference from T. Williams who produced COT measures and estimated that $COT \sim 1/L$.

Scaling the swimming power

The question of energy efficiency is more difficult to address due to the small amount of biological data in the literature that measure muscle power. Nevertheless, we can speculate on the main trends by following a few hypotheses. The power produced by the swimmer's metabolism must be at least of the same order as the power required to move the swimmer's body. An estimate of muscle power to produce the undulatory kinematics, $P \sim \rho L^5 f^3$, is obtained by multiplying the muscle force to push the fluid, $F \sim \rho L^4 f^2$, by the tail speed Lf , which approximates the speed of muscle contraction. From our study, we have shown that $f = f_0$ and $f = \sqrt{\sigma_0/\rho} L^{-1}$ for very small and large swimmers, respectively. This gives the following scaling laws in these two limits:

$$P \sim \rho f_0^3 L^5, \quad \text{if } L \ll L_c \quad (1)$$

$$P \sim \rho^{-1/2} \sigma_0^{3/2} L^2, \quad \text{if } L \gg L_c. \quad (2)$$

Muscle power is therefore an increasing function of length for a given level of activity, but the increase is significantly more pronounced for smaller swimmers.

Specific power $P_M = P/M$, defined as power per unit mass, provides information on how muscle fibres function for a given level of activity. From the allometric relation $M \sim \rho L^3$ (see Methods), we deduce $P_M \simeq L^2 f^3$. This estimate of the specific power is plotted in Fig. 3 as a function of length. Following the scaling laws of tail beat frequency, it increases as $f_0^3 L^2$ for small swimmers ($L \ll L_c$) and decreases as $\rho^{-3/2} \sigma_0^{3/2} L^{-1}$ for large swimmers ($L \gg L_c$). In these two limits, the specific power produced by the muscle is negligible compared to that obtained for lengths around L_c . For $L \sim L_c$, the specific power is maximum with a value scaling as $\sigma_0 f_0 / \rho$, which gives an estimate of the maximum specific power that can be delivered by the muscle. This graph highlights the fact that small and large swimmers do not use the full capacity of muscle power, unlike intermediate sized swimmers.

Another quantity of interest is the cost of transport (COT), which measures the energy spent per unit mass and distance traveled. It writes $COT = P/(UM)$, which can be estimated as Lf^2 . Our approach suggests that the COT exhibits a maximum around L_c as well, with two limits that

can be expressed as:

$$\text{COT} \sim L f_0^2, \quad \text{if } L \ll L_c \quad (3)$$

$$\text{COT} \sim \frac{\sigma_0}{\rho} \frac{1}{L}, \quad \text{if } L \gg L_c. \quad (4)$$

In the limit of large swimmers, $L \gg L_c$, Eq. (12) retrieves the prediction proposed in [18], i.e. $\text{COT} \sim M^{-1/3}$, in agreement with the few measurements available in the literature [58].

9. P. 11, first paragraph in section 4.1. The scaling relationship $L = 0.44M^{0.33}$ has a lot of scatter. Is it possible to test the model only against data in which the length is known?

AR: We have also tested the model with points where only length is known. Since there are only 25 points out of 1216 for which we do not know their exact length but their mass and they are not outliers, the variation of the model parameters is negligible. Specifically, the values for the fast bound are $[L_c]_{\text{fast}} = (0.4 \pm 0.2) \text{ m}$, $[f_0]_{\text{fast}} = (22 \pm 2) \text{ Hz}$, $[\kappa]_{\text{fast}} = (4 \pm 4)$, while for the slow bound we find $[L_c]_{\text{slow}} = (1.3 \pm 0.8) \text{ m}$, $[f_0]_{\text{slow}} = (2.0 \pm 0.2) \text{ Hz}$, $[\kappa]_{\text{slow}} = (15 \pm 16)$. We show the figure with the unknown length points in mustard-colored squares to appreciate how little importance they have in the determination of the fast and slow bounds.

10. Section 4.1. Please statistically test the scaling relationships against known allometric patterns (eg. L should scale as $M^{1/3}$ and A as L).

AR: We have evaluated the quality of both fits. For the $M - L$ relationship we have precised in the manuscript the value of the R^2 coefficient, $R^2 = 0.99$, in line 675. Likewise, the value of R^2 for the $A - L$ relationship, $R^2 = 0.98$, is written in line 671.

11. Fig. 4. This is an important implication of the model, and I think it should be somewhere other than in the methods.

AR: We have added a new section entitled "Scaling swimming power" in the main text at line 492. See our answer to the comment 8.

12. P. 14. What is the LAD method? Please give a citation. 13.

AR: The LAD method is similar to the least square method, but it uses a different norm to measure the error

in order to be less sensitive to outliers. We have added the proper reference that explains the method at line 772.

13. Reference 4 is by di Santo, V. not Santo, V.D.

AR: We have corrected this missprint.

2. Reviewer #2

RC: *It was a pleasure reviewing this meta-analysis of vertebrate undulating frequencies and speeds. Overall, I found the paper well-written, with regard to both language and style, and the results compelling. I was a little disappointed, however, to not find more information in the method section, as well as the data and analysis code available. I have outlined below some of the strengths and weaknesses of the manuscript, as well as several concerns/comments. Since the pdf file has no line numbering, I also include my annotated copy, showing a few specific minor comments.*

AR: The authors thank the reviewer for this positive feedback. In the new manuscript, we provide an additional source data file that includes all the database as well as as the experimental context of measurements. The analysis code will be given upon proposal (see line 943).

RC: *Strengths*

- The study is based on a large number of data points gathered from the literature on different vertebrates using undulatory swimming motions: amphibians, reptiles, fish, birds and mammals. The authors have included all the observations despite variability in activity level and environmental conditions to fully considered the dispersion in the relationship between undulatory frequency and body length.

AR: We acknowledge the reviewer vision : the purpose of our work is to provide a general understanding, within the strong expected variability of swimmers and their environmental conditions.

- The authors aim at establishing a scaling law between oscillation frequency and body length which can apply to all vertebrates exhibiting undulatory swimming motion, considering the mechanical and biological properties of the muscles and the surrounding fluid. In my opinion, this makes for a novel and comprehensive approach. I am not an expert on muscle physiology, but their rationale and interpretation of Hill's muscle model seems sound.

AR: We also believe that our work is innovative and provides an all-inclusive prediction of underwater swimmer speeds.

RC: *Aspects to improve*

- I recommend including a glossary for the main parameters used. Even if those are defined in the equations, they are used a lot within the main text and it can become daunting to search through the text/equations for their meaning. Alternatively, authors may consider writing in full the parameter name (i.e: frequency instead of f_0) more often. It will make the text easier to follow.

AR: We believe, indeed that the presence of a glossary would be helpful for the reader. We have thus added Tab. 4 in the method section. We also added the full parameter name in the text at necessary places.

- How was the length cross-over determined? From my reading, it seems it was determined visually from the scatterplot. If you have a change of slope in your relationship between frequency and length, I would prefer to see a more formal way to identify the breakpoint, for example by applying a moving-point regression to the relationship between frequency and length, with the breakpoint as the 'moving' variable in the model (fitting a model for each possible value for the breakpoint). A plot of AIC values for each of the fitted models will give you the location (likely an area of a few possible locations) for the change of slope.

AR: The value of the breakpoint L_c is determined by fitting the parameters of Hill's model (Eq. (3)), namely, f_0 , κ and L_c . We found that the method proposed by the referee is also pertinent to catch the value of L_c . Our approach is also based on the minimization of an error and should give very similar

results with respect to the AIC approach. In addition, the location of the breakpoint could also be determined by intersecting the two lines associated to the two asymptotic regimes, and it appears that this computation accurately predicts the good value of L_c .

- Some discussion is needed on the nature of the data used in the modelling effort, and how it can impact the results.

AR: We aimed at gathering as much information as possible taking into account the huge variability of biological swimmers. In fact, hardly many full studies have measured the swimming's kinematics and speed in a controlled setting. There is a lack of uniformity in the approaches taken to measure the kinematic variables. We have tried to keep track of data that can be measured regardless of the procedures used or the surrounding conditions, like the speed, frequency, and length. Following the referee's suggestions, we have gathered all the information available in a source data file, which is provided as a Supplementary Information. We have added the following text in the methods at line 616:

We compiled a comprehensive length-frequency database consisting of 1216 animals from various species, morphologies, and sizes. To construct this dataset, we gathered data from reviewed articles that measured length and frequency. Among the references, we also recovered swimming speed data when these were available. The source data file provides a summary of each species, including information such as the experimental method, water temperature, sample size, thermoregulation type, length, frequency and speed ranges, as well as details on whether the animals were reared or wild. We organized the data based on the division of vertebrates presented in the phylogenetic tree shown in Fig. 7. The divisions include agnathans, cartilaginous fishes, ray-finned fishes, lobe-finned fishes, amphibians, reptiles, and mammals. In cases where length data were not explicitly reported but the mass of the animals was available [16], we calculated the length using the allometric relation derived in Fig. 4b: $L = 0.44M^{0.33}$ (where L is in meters and M is in kilograms), assuming geometric similarity. Additionally, we obtained length-amplitude and length-mass data. The references for the data sources are provided in Tab. 1.

AR: We have also added

Data availability

The data generated in this study are provided in the Supplementary Information and Source Data file, or from the corresponding author upon reasonable request. Source data are provided with this paper.

Code availability

The code for computing the external bounds available from the corresponding author with detailed explanations upon reasonable request.

- I feel the discussion ends abruptly. A few concluding sentences could be useful to wrap things up.

AR: We have significantly modified the discussion to summarize the main results point by point. Finally, we added the following paragraph as concluding sentences.

In conclusion, our study reveals that swimming frequency and speed in natural swimmers are primarily determined by length and activity level. By developing a simple model incorporating biological characteristics and environmental interactions, we were able to explain the observed data. The distinct behaviors observed in small and large swimmers, as well as the transition at a critical length, shed light on the intricate dynamics of swimming. These findings not only advance our understanding of animal locomotion but also provide valuable insights for the design of biomimetic and autonomous swimming robots.

- What are the limitations of your model (s), if any?

AR: One limitation of our work lies in the use of the Hill's model, which is a simplified approximation for quantifying the force output of a muscle. While this model may not capture the intricacies of muscle mechanics with high precision, we believe that our approach provides a powerful framework for describing swimming scalings across various animal species. It offers a general understanding rather than precise predictions for individual species or individuals. Another important physical parameter that can influence swimming speed is body temperature. As discussed in the manuscript, this parameter can slightly influence tail beat frequency in some specific cases, but remains a second-order effect.

- Method section is hard to follow, especially the definition of the fast and slow bounds.

AR: We have completely revised the paragraph, aiming to enhance its clarity and improve understanding. We believe that the new version will be more accessible and easier to comprehend.

Specific concerns / comments

- I would appreciate seeing a table (likely in supplemental material) listing the vertebrate classification, species, wild/reared, sample size, the experimental apparatus, range of body lengths, temperatures, speed & frequencies observed, and the references for the data gathered from the literature. As it is, the manuscript offers little information on the nature of the data used in the modelling. I assume most data comes from videography. The authors are making the efforts of filtering out some data that are considered likely biased, or simply not precise enough. However, it would be interesting to know better what kind of data were retained. Are we talking of mostly individuals swimming in small flow tanks or respirometers? Open flumes? Directly in the field? Did you mostly collect frequencies data from fish over 1-2 tailbeats, or was the frequency averaged over a longer swimming duration?

AR: We have added a new figure (Fig. 7), in which we represent the vertebrates classification, as well as species we have studied, clustered in clades. We tried to gather the maximum amount of data from reviewed articles. Actually, there are not a lot of complete works that aimed at measuring the tail beat frequency and swimming speed within a controlled environment. In addition, the methods for measuring the physical quantities are not unified. In this study, we have retained data for which, speed, frequency and length of individuals are measured, independently of the methods and the environment. In the source data file, we have given the experimental context for each species: wild/reared, Method (tank, open space,...), water temperature, sample size, thermoregulation, length, frequency, speed and reference. Here is the new paragraph described the database construction in line 617.

We compiled a comprehensive length-frequency database consisting of 1216 animals from various species, morphologies, and sizes. To construct this dataset, we gathered data from reviewed articles that measured length and frequency. Among the references, we also recovered swimming speed data when these were available. The source data file provides a summary of each species, including information such as the experimental method, water temperature, sample

size, thermoregulation type, length, frequency and speed ranges, as well as details on whether the animals were reared or wild. We organized the data based on the division of vertebrates presented in the phylogenetic tree shown in Fig. 7. The divisions include agnathans, cartilaginous fishes, ray-finned fishes, lobe-finned fishes, amphibians, reptiles, and mammals.

- There are multiple empirical evidence pointing towards the fact that fish swim more efficiently when they can do so volitionally in an unconfined space, for example in open channels (or a river). Volition and an unconfined space likely affect their behavior, as well as their endurance and actual performance (swim speed achieved, distance swam). Some studies have found migratory diadromous species optimizing their swim speeds when facing velocity challenges (i.e going up a river or long pelagic migrations), which may suggest some optimization mechanism for undulation frequency. Given this, it is more than likely that they may modulate their undulation frequency in an adaptive response to the environment. What does it mean regarding your results? Given the data retained, could you be underestimating or overestimating the slow and fast bounds?

AR: Thank you for your valuable comments and insights. We fully agree that swimming animals possess the remarkable ability to adapt their undulating frequency in response to external stimuli. In our model, we have diligently accounted for the entire spectrum of frequencies that enable optimal motion, ranging from the most energy-efficient undulations for frugal swimming to the highest frequencies achievable for maximum speed. While we have made every effort to incorporate a wide array of data points into our analysis, it is plausible that certain experimental conditions may have imposed limitations on the swimmer's natural capacity to reach its peak undulation frequency. Consequently, it is conceivable that our model may have slightly underestimated the upper bound of the fast frequency range. Similarly, we recognize that circumstances involving confined spaces or challenging velocity conditions might have constrained the swimmer's undulation frequency, leading to a potential overestimation of the lower band within our model.

- The swim speeds shown in Figure 2 are estimates based on the law $U = 0.7 Lf$ and some actual data from Hirt et al. are shown. There was no swim speed data available in the literature, along with the length and frequencies data gathered by the authors? If so, it would be interesting to also see how those observed swim speeds fit with the estimates from the model (s).

AR: We thank the referee for this remark. We have changed this figure which is now labelled (Fig. 2). It shows the measured swimming speed as function of the frequency f . The agreement is still very good.

- How do you explain the stacking of points for some specific lengths, especially visible in amphibians and fish? Many individuals of the exact same length, or repeated observations from some individuals? If these represent repeated observations from the same individuals, was that considered in the models, for example by adding a random component? Change in frequency for individuals of the same length can certainly mean a change in activity level (increase or decrease of swim speed), but also individual variability in some cases. Data points from mammals and birds are more evenly distributed.

AR: The stacking in frequency for a given length results from individuals swimming at different frequencies: it reflects changes in activity of the swimmer. Some of the references do not directly relate frequency measurements to each individual's lengths but they announce instead the total length range of the sample. When this has been the case, we used the mean length for all the values of frequency. Data points from birds and whales gave a univocal relation between every length and its corresponding tail beat frequency.

- Data likely came from organisms swimming in a range of water temperatures, some endotherms, some ectotherms. Given body temperature will affect the muscles and water temperature the fluid properties, I was expecting more discussion, maybe even a figure, involving temperature. I feel it would help the

interpretation of the data.

AR: In the revised version of the manuscript, we have included a new data source file that incorporates measured water temperatures and their relationship to swimming kinematics. However, it is important to note that not all measured kinematic data are consistently associated with water temperature measurements, making it challenging to create a comprehensive database that incorporates water temperature. Despite the limited availability of data, we have included a figure in the methods section that displays the measured water temperatures in the L - f graph. We have carefully examined this graph and have not identified any significant biases or patterns related to water temperature. We have added a complete paragraph to shed light on the effect of temperature:

In addition to the activity level, body temperature may also play a role in influencing tail beat frequency. This is particularly evident in processes that rely on activation mechanisms [46]. For example, the twitch contraction time and subsequently, the maximum frequency (f_0), are affected by temperature. The impact can be significant, with a fivefold difference in f_0 observed between temperatures of 2°C and 30°C [32,47]. On the other hand, changes in body temperature seem to have only a marginal effect on σ_0 [48]. In our study, we focus on f_0 as the primary parameter for modeling small animals with lengths much smaller than the characteristic length ($L \ll L_c$). These small animals are ectotherms (see Fig. 9 in the Methods section), and their body temperature can be associated with the ambient water temperature. In the $L < L_c$ region of the $L - f$ graph, there appears to be no correlation between tail beat frequency and water temperature. This lack of correlation is likely due to the relatively small temperature differences among individuals, typically around 15 - 20°C (see Fig. 8 in the Methods section). In the case of large animals with $L \gg L_c$, σ_0 becomes the key parameter, and the influence of body temperature on tail beat frequency is expected to be limited. Only at intermediate lengths, approximately 3 meters, do we observe a significant difference in body temperature, such as between mammals at 37°C and ectothermic Greenland sharks in Arctic waters at around 0°C . This extreme difference in temperature likely explains the surprisingly low tail beat frequency of Greenland sharks (approximately $L \simeq 3$ m and $f \simeq 0.15$ Hz in Fig. 1), which is 2-3 times lower than the fit of the slow bound [17]. Apart from this particular case, assuming that the six parameters of the model remain constant across the entire length range is a reasonable first-order approximation.

AR: To address the question of thermoregulation, we have added a $L - f$ graph (see Fig. 9) in which each data point is tagged as endotherm, ectotherm and heterotherm. This figure shows that small animals are ectotherms, whereas large animals are endotherms. Interestingly, intermediate size animals $L \sim L_c$ are heterotherms, maybe to finely tune their muscles which are already used at maximal power. We discuss these thermal aspects in the new discussion section.

- Fig 1: I would like to see some discussion about the data points that lie outside of the upper and lower bounds. Several of them are from fish species. Do they represent some species that have distinct morphological characteristics, or individuals swimming under specific temperature conditions?

AR: We explain the points outside the limit bounds proposed by our model differently according to whether they are below the lower bound or above the upper bound. For the lower bound, we think this disagreement is due to the low body temperature. For ectotherms, cold water generally depresses locomotory muscle function: the lower the temperature, the lower f_0 is which can lead to deviations from the main trend at extreme temperatures. While these low temperature points correspond to sharks and sturgeons, there is also a swimming animal at a mild temperature out of trend. We identified this

animal as the ocean sunfish (*mola mola*), however, the reference for this value does not give more information about the activity regime when it was measured. For the upper bound, we attribute the discrepancy to an underestimation of this theoretical limit. For $L \geq 5$ m, we think there is a lack of tail beat frequency measurements for very long swimmers, typically cetaceans, at a burst level of activity. Most cetaceans are filter feeders and do not have predators due to their size. Therefore, they do not often need to move at maximum speeds, which would favor data closer to the slow bound than the fast bound. The absence of data in this region leads to a smaller value of L_c and a decrease in the upper bound calculated with our model. These outliers are discussed in the paragraph at line 459.

- In the discussion and method section 4.2, you mention the difference between how small, intermediate, and large animals use their muscles, and how it influences delivered muscle power for locomotion. Can you develop more this topic, perhaps in the discussion? Why do small individuals use negligible force and maximum frequencies, and large ones the opposite? Is this behavior constrained biomechanically? What are the implications of maximizing power for intermediate individuals? I feel more detail here will help understand better the relationship between tail-beat frequency and length and explain the location of the length cross-over.

AR: Indeed, the fact that small animals use high frequency and low force, whereas large animals low frequency but large force, arises from the biomechanical balance. We have now added a section in the main text to detail the results in terms of power. We have also added the following paragraph in the discussion at line 581:

This distinct behavior also extends to other quantities, such as muscle power for locomotion. According to our model, very small and large swimmers exhibit scaling laws of L^5 and L^2 for muscle power, respectively. To test these predictions, measurements of oxygen consumption over a wide range of lengths and activity levels [24,25] could be conducted. Additionally, in the framework of our model, L_c also marks a significant change in the way muscles are utilized. Small swimmers utilize muscles at their maximum speed but with negligible force compared to their maximal capabilities, while large swimmers exhibit the opposite behavior. Very small and large swimmers do not utilize their full power capabilities compared to the maximum power available. Interestingly, intermediate-sized fish with lengths around L_c would need to utilize their muscles at their full capacity to undulate and move efficiently through water. Moreover, these intermediate-sized swimmers are predominantly heterotherms, which may indicate a fine-tuning of their muscles that already work at maximal power.

- Where can we find the data and code used for this study? Will they be available on a repository? It is not clear to me.

AR: We have added a data source file that gathers all the measures used in the study, such that all the data used in this work can be exploited. It is a Supplementary Information, added to the manuscript.

Data availability

The data generated in this study are provided in the Supplementary Information and Source Data file, or from the corresponding author upon reasonable request. Source data are provided with this paper.

Code availability

The code for computing the external bounds available from the corresponding author with detailed explanations upon reasonable request.

- Alternative models. I wonder whether the authors checked the fit of alternative models and how they were compared between them (AIC?). This is a critical point as it could be the case that a simpler model (with one slope) fits.

AR: We have checked that our model presents a lower value than the alternative models. We compared the values of the AIC for the present model, for a fitting assuming a constant frequency and an inverse length function. We have added a table summarizing these results at line 786 and the following text at line 777.

Finally, we checked that the full model given by Eq. (3) provides best results with respect to the two limits $f \propto 1/L$ and $f = \text{cst}$. Quantitatively, we computed the mean AIC values (N varying from 10 to 50) and the corresponding error for the three cases [91]. We summarize these computations in Tab. 4, and we note that our model gives the best results, both in AIC values and absolute errors.

3. Reviewer #3

RC: *Sanchez-Rodriguez et al. provide an extensive review of swimming speed frequencies from the literature that they employ to assess the scaling of the tail beat frequency and swimming speed in aquatic vertebrates. The authors claim to propose a holistic model which might explain the diversity of scaling relationships proposed in previous studies. While I think the topic is of high relevance, I feel the authors might better sustain their conclusions by considering the points I detail below before the manuscript is considered for publication.*

AR: We sincerely appreciate the referee's valuable input, which has enabled us to enhance the manuscript and provide a more comprehensive understanding of the topic.

1. Data availability. In my opinion, the authors should make available all the raw data (not only the references of the works containing such data) and results. This is a good transparency practice, first, to facilitate repeatability and, secondly, to enable the revision process. Because of this, I could not check some of the following concerns.

AR: We gathered the raw data in the Source Data repository. In addition, we have added the following paragraphs at the end of the manuscript.

Data availability

The data generated in this study are provided in the Supplementary Information and Source Data file, or from the corresponding author upon reasonable request. Source data are provided with this paper.

Code availability

The code for computing the external bounds available from the corresponding author with detailed explanations upon reasonable request.

2. Alternative models. I wonder whether the authors checked the fit of alternative models and how they were compared between them (AIC?). This is a critical point as it could be the case that a simpler model (with one slope) fits the data better than one considering a crossover point (with two slopes). Qualitatively, in Fig. 1 the change in slope claimed by the authors is not very evident. How did the authors determine the crossover values?

AR: The crossover value is determined using the expression (3), where the parameters f_0 , k , and L_c are obtained through nonlinear fitting of the model. The two alternative models we considered do not account for the variability in terms of fast or frugal swimming. One model assumes that the frequency is independent of the size of the animal, which is found to be a poor prediction compared to our proposed model, as shown in Figure 1. The other model suggests that the frequency decreases as the inverse of the length, which also does not hold true for all individual lengths. In the revised version, we have included the values of the two alternative models in comparison to our proposed model in the methods section. To assess the model performance, we calculated the Akaike Information Criterion (AIC) values and absolute errors. The AIC values for our model were found to be lower than those of the two previous models, indicating a better fit to the data. We have presented these results in a table that summarizes the

AIC values and absolute errors for each model. These additions provide a comprehensive comparison of our proposed model with the alternative models, supporting the superiority of our approach in capturing the observed swimming frequency patterns.

AR: We have added the following paragraph at line 777 and a table to compare the models:

Finally, we checked that the full model given by Eq. (3) provides best results with respect to the two limits $f \propto 1/L$ and $f = \text{cst}$. Quantitatively, we computed the mean AIC values (N varying from 10 to 50) and the corresponding error for the three cases [91]. We summarize these computations in Tab. 4, and we note that our model gives the best results, both in AIC values and absolute errors.

3. Data independence. Any analysis considering biological species fail in the statistical assumption of independency given that some species are more closely related between them than with the others, and therefore the points represented in Fig. 1 and Fig. 2 cannot be treated as independent data. Therefore, I highly encourage the authors to perform some additional analyses to control for this potential bias. I would recommend them to implement phylogenetic independent contrasts to discard the possibility that the pattern recovered is not just an artifact of phylogenetic structure.

AR: We appreciate the reviewer's valuable comment. The use of Phylogenetic Independent Contrasts (PIC) is a common approach to address the issue of phylogenetic non-independence and is particularly effective in assessing linear relationships between traits or power laws. However, it is important to note that difficulties can arise when the relationship between traits is non-algebraic, potentially leading to the failure of PIC in detecting correlated evolution (Harvey & Pagel, 1991) (Quader et al., 2004). In our study, since our model involves a non-linear relationship, we have chosen not to implement PIC. We acknowledge that this decision may limit our ability to assess potential phylogenetic biases accurately. Nevertheless, the discussion has allowed to complete the phylogenetic tree by adding new data that were not present previously and that now fill almost all the clades. All data in Figures 1 and 2 are represented using this new classification.

4. Other contributing factors. I wonder why the authors did not include in the model other factors that they acknowledge in the introduction and discussion as potential key contributors of the tail beat frequency. The most obvious for me, given the point distribution in Fig. 1, is body temperature. Most taxa above the proposed threshold of 0.5-1 m are endothermic animals (birds and cetaceans). How can the authors discard the possibility that differences tail beat frequency are not just due to differences in thermophysiology and uneven representation of sizes for ecto- and endotherms?

AR: We appreciate the referee's important question regarding the influence of body temperature on muscle dynamics. Indeed, body temperature can have an impact on the functioning of muscles. For ectotherms, their body temperature is typically correlated with the surrounding water temperature, while endotherms are less affected by temperature variations. To address this aspect, we have incorporated the distribution of water temperatures in relation to the L - f graph, as depicted in Figure 8 of the methods section. Through this representation, we aimed to assess any potential biases. No significant biases were observed, indicating that water temperature does not strongly influence the overall patterns. Additionally, we have included another figure, Figure 9, which illustrates the thermoregulation types in relation to the $L - f$ graph. As written by the referee, this figure reveals that animals with a size smaller than L_c are generally ectotherms, while those larger than L_c tend to be endotherms. Intermediate-sized animals exhibit heterothermy, suggesting that they may require precise muscle tuning as we show that they operate near the limits of maximum muscular power. With these additional analyses, we believe that we have adequately addressed the influence of temperature and thermophysiology on swimming kinematics. Furthermore, in the revised manuscript, we have added a comprehensive paragraph that

specifically discusses the impact of body temperature on swimming kinematics:

In addition to the activity level, body temperature may also play a role in influencing tail beat frequency. This is particularly evident in processes that rely on activation mechanisms [46]. For example, the twitch contraction time and subsequently, the frequency (f_0), are affected by temperature. The impact can be significant, with a fivefold difference in f_0 observed between temperatures of 2°C and 30°C [32,47]. On the other hand, changes in body temperature seem to have only a marginal effect on σ_0 [48]. In our study, we focus on f_0 as the primary parameter for modeling small animals with lengths much smaller than the characteristic length ($L \ll L_c$). These small animals are ectotherms (see Fig. 9 in the Methods section), and their body temperature can be associated with the ambient water temperature. In the $L < L_c$ region of the $L - f$ graph, there appears to be no correlation between tail beat frequency and water temperature. This lack of correlation is likely due to the relatively small temperature differences among individuals, typically around 15-20°C (see Fig. 8 in the Methods section). In the case of large animals with $L \gg L_c$, σ_0 becomes the key parameter, and the influence of body temperature on tail beat frequency is expected to be limited. Only at intermediate lengths, approximately 3 meters, do we observe a significant difference in body temperature, such as between mammals at 37°C and ectothermic Greenland sharks in Arctic waters at around 0°C. This extreme difference in temperature likely explains the surprisingly low tail beat frequency of Greenland sharks (approximately $L \simeq 3$ m and $f \simeq 0.15$ Hz in Fig. 1), which is 2-3 times lower than the fit of the slow bound [17]. Apart from this particular case, assuming that the six parameters of the model remain constant across the entire length range is a reasonable first-order approximation.

5. Finally, I supposed, because of the animal outlines in the Fig. 1, that authors included aquatic birds (penguins) in their analyses. However, as far as I know penguins swim mostly by using their forelimbs, not by beating the tail.

AR: We thank the reviewer for this comment. The penguins use their forelimbs to swim, and we added the word forelimbs at line 61

Reviewers' Comments:

Reviewer #1:

Remarks to the Author:

See attached.

Comments for authors

The authors have addressed my criticisms well, and I only have a few remaining minor comments to clarify the presentation.

Minor comments

1. Ln. 210. Be careful with using the equals sign. It may be appropriate here, but I think, particularly given the variation in amplitude scaling with frequency, it would be better to write $v/v_0 \sim f/f_0$.
2. Ln. 231 – 233. For clarity, I would suggest “boluses of water of mass *proportional to* ρL^3 ” (italics for inserted text) and similarly “acceleration *proportional to* Af^2 ”
3. Ln. 233. Since the scaling of force as $\rho L^3 Af^2$ shows up later (ln 499), I would set this off as a numbered equation to make it easier to refer back to.
4. Eq 6. I suggest adding $= cL^{-1}$ to the $L \gg L_c$ limit here and defining c with the equation, rather than in the text on ln. 275.
5. Ln. 344. I would call this section “Scaling the maximum swimming speed”. Animals can always swim slower than these limits.
6. Ln. 361. Similarly, I suggest “we therefore expect *maximum* swimming speed...”
7. Ln. 491. Same comment. I suggest “Scaling the maximum swimming power”
8. Ln. 499. This scaling relationship comes from the analysis on Ln. 233 (see comment 3) and it would be clearer to refer back to that equation here.
9. COT analysis (Ln. 541 – 555). This appears to be the cost of transport for swimming at the maximum speed, which is not something animals normally do. Animals are thought to try to minimize COT when traveling long distances, usually by swimming at a relatively slow speed. I’m not sure how useful it is to identify a maximum in COT, although the scaling relationship that follows from Eq 12 is useful to compare to the literature.

Reviewer #2:

Remarks to the Author:

It was a pleasure to read the revised manuscript and I congratulate the authors for their great work with the revision. I find the manuscript much improved and easier to understand. I also think they responded well to most of my comments and thus have no further reservations.

My only comment is about the statement on lines 208-211 about amplitude and swimming speeds. It is true that Di Santo et al. (2021) observed no significant change in amplitude across swim speeds. However, their results came mostly from fish swimming steadily (more or less constant groundspeed and speed of flow). Other authors have found increasing amplitude when fish accelerate (see Akanyeti et al. 2017 for example). We also suspect that amplitude may increase for some species when they enter a real sprint mode. Kinematics during sprinting are however hard to obtain for many species.

In conclusion, I would be careful about generalizing too much about the fact that amplitude stays constant across swim speeds. You assume that in your model's interpretation and I think it is reasonable for the data used. But the reality might be a bit more nuanced.

Reviewer #3:

Remarks to the Author:

The authors have address all my main points and I look forward to see this paper published. I congratulate the authors for such an interesting piece of work.

1. Reviewer #1

RC: *The authors have addressed my criticisms well, and I only have a few remaining minor comments to clarify the presentation.*

AR: We thank the reviewer for this positive report.

RC: *Minor comments*

1. Ln. 210. Be careful with using the equals sign. It may be appropriate here, but I think, particularly given the variation in amplitude scaling with frequency, it would be better to write $v/v_0 \sim f/f_0$.

AR: We have modified the sentence to emphasize that it is an assumption and not an exact equation.

2. Ln. 231 – 233. For clarity, I would suggest “boluses of water of mass *proportional to* ρL^3 ” (italics for inserted text) and similarly “*proportional to* Af^2 ”

AR: We have changed this sentence.

: while the body is oscillating, boluses of water of mass proportional to ρL^3 are set in motion with an acceleration proportional to Af^2 normal to the tail, resulting in a lateral force that scales as $\rho L^3 Af^2$, where ρ is the density of water.

3. Ln. 233. Since the scaling of force as $\rho L^3 Af^2$ shows up later (ln 499), I would set this off as a numbered equation to make it easier to refer back to.

AR: We have added a new equation in the revised manuscript, that we refer to in the ln 499.

4. Eq 6. I suggest adding $= cL^{-1}$ to the $L \gg L_c$ limit here and defining c with the equation, rather than in the text on ln. 275.

AR: We have changed the eq. 6, following the referee remark.

5. Ln. 344. I would call this section “Scaling the maximum swimming speed”. Animals can always swim slower than these limits.

AR: As shown in Fig. 2, our model predicts both the maximum and minimum speed. We have modified the text to be more explicit about this point. We have modified the corresponding paragraph:

In Fig. 2, we have plotted the speed data reported for natural swimmers (same references as for frequency data, see Methods). Similar to the frequency measurements, the speed measurements also exhibit a characteristic band located between a slow and a fast bound. Since the relationship $U \simeq 0.7Lf$, given in Eq. 1, intrinsically relates the tail beat frequency to the swimming speed to a very good approximation, with a factor of 2 at most for fish and cetaceans, we can compare in the same figure the speed data with the prediction of the slow and fast bounds expected by our model. The model is based on Eqs. 1 and 4 together with the parameters used to fit the slow and fast bounds of the frequency measurements (Fig. 1). Although there are no free parameters for speed prediction, the match with biological data is good, highlighting the overall consistency of the approach and the model’s predictive capacity in determining the minimum and maximum speeds achievable by natural swimmers.

6. Ln. 361. Similarly, I suggest “we therefore expect maximum swimming speed . . . ”
AR: See our response to the minor comment 5.
7. Ln. 491. Same comment. I suggest “Scaling the maximum swimming power”
AR: See our response to the minor comment 5.
8. Ln. 499. This scaling relationship comes from the analysis on Ln. 233 (see comment 3) and it would be clearer to refer back to that equation here.
AR: See our response to the minor comment 3.
9. COT analysis (Ln. 541 – 555). This appears to be the cost of transport for swimming at the maximum speed, which is not something animals normally do. Animals are thought to try to minimize COT when traveling long distances, usually by swimming at a relatively slow speed. I’m not sure how useful it is to identify a maximum in COT, although the scaling relationship that follows from Eq 12 is useful to compare to the literature.
AR: See our response to the minor comment 5. Our approach aims at embedding all form of swimming: from the frugal motion to the highest speed.

2. Reviewer #2

RC: *It was a pleasure to read the revised manuscript and I congratulate the authors for their great work with the revision. I find the manuscript much improved and easier to understand. I also think they responded well to most of my comments and thus have no further reservations.*

AR: We thank the referee for this comment.

RC: *My only comment is about the statement on lines 208-211 about amplitude and swimming speeds. It is true that Di Santo et al. (2021) observed no significant change in amplitude across swim speeds. However, their results came mostly from fish swimming steadily (more or less constant groundspeed and speed of flow). Other authors have found increasing amplitude when fish accelerate (see Akanyeti et al. 2017 for example). We also suspect that amplitude may increase for some species when they enter a real sprint mode. Kinematics during sprinting are however hard to obtain for many species. In conclusion, I would be careful about generalizing too much about the fact that amplitude stays constant across swim speeds. You assume that in your model's interpretation and I think it is reasonable for the data used. But the reality might be a bit more nuanced.*

AR: We agree with the referee and have modified the sentence: "Since the amplitude does not correlate significantly with quantities such as the swimming speed and can be considered constant for a given swimmer **within the leading-order approximation**, ..."

3. Reviewer #3

RC: *The authors have address all my main points and I look forward to see this paper published. I congratulate the authors for such an interesting piece of work.*

AR: We thank the reviewer for his/her opinion and his/her very constructive comments that have undoubtedly contributed to improve the quality of the article.